COMMUNICATIONS

# Anatomical dissociation of intracerebral signals for reward and punishment prediction errors in humans

Maëlle C. M. Gueguen [1], Alizée Lopez-Persem [2], Pablo Billeke [3], Jean-Philippe Lachaux[4], Sylvain Rheims [5], Philippe Kahane[6], Lorella Minotti[6], Olivier David[1], Mathias Pessiglione [7,8,9] & Julien Bastin [1,9 ✉]

Whether maximizing rewards and minimizing punishments rely on distinct brain systems remains debated, given inconsistent results coming from human neuroimaging and animal electrophysiology studies. Bridging the gap across techniques, we recorded intracerebral activity from twenty participants while they performed an instrumental learning task. We found that both reward and punishment prediction errors (PE), estimated from computational modeling of choice behavior, correlate positively with broadband gamma activity (BGA) in several brain regions. In all cases, BGA scaled positively with the outcome (reward or punishment versus nothing) and negatively with the expectation (predictability of reward or punishment). However, reward PE were better signaled in some regions (such as the ventromedial prefrontal and lateral orbitofrontal cortex), and punishment PE in other regions (such as the anterior insula and dorsolateral prefrontal cortex). These regions might therefore belong to brain systems that differentially contribute to the repetition of rewarded choices and the avoidance of punished choices.

[1] Univ. Grenoble Alpes, Inserm, U1216, Grenoble Institut Neurosciences, GIN, Grenoble, France. [2] Frontal Function and Pathology team, Institut du Cerveau, Sorbonne Université, INSERM U 1127, CNRS UMR 7225, Paris, France. [3] División de Neurociencia, Centro de Investigación en Complejidad Social (neuroCICS), Facultad de Gobierno, Universidad del Desarrollo, Santiago, Chile. [4] Lyon Neuroscience Research Center, Brain Dynamics and Cognition team, DYCOG INSERM UMRS 1028, CNRS UMR 5292, Université de Lyon, Lyon, France. [5] Department of Functional Neurology and Epileptology, Hospices Civils de Lyon and University of Lyon, Lyon, France. [6] Univ. Grenoble Alpes, Inserm, U1216, CHU Grenoble Alpes, Grenoble Institut Neurosciences, GIN, Grenoble, France. [7] Motivation, Brain and Behavior lab, Centre de Neuroimagerie de Recherche, Institut du Cerveau et de la Moelle épinière, Hôpital de la Pitié-Salpêtrière, Paris, France. [8] Inserm U1127, CNRS U7225, Université Pierre et Marie Curie (UPMC-Paris 6), Paris, France. [9] These authors contributed equally: Mathias Pessiglione, Julien Bastin. ✉email: julien.bastin@univ-grenoble-alpes.fr

Approaching reward and avoiding punishment are the two fundamental drives of animal behavior. In principle, both reward-seeking and punishment-avoidance could be learned through the same algorithmic steps. One the most straight and simple algorithm postulates that the value of chosen action is updated in proportion to prediction error[1,2], defined as observed minus expected outcome value. In this simple reinforcement learning model, the only difference is outcome valence: positive for reward (increasing action value) and negative for punishment (decreasing action value). The same brain machinery could therefore implement both reward and punishment learning.

Yet, different lines of evidence point to an anatomic divide between reward and punishment learning systems, in relation with opponent approach and avoidance motor behaviors[3,4]. First, fMRI studies have located prediction error (PE) signals in different brain regions, such as the ventral striatum and ventromedial prefrontal cortex (vmPFC) for reward versus the amygdala, anterior insula (aINS), or lateral orbitofrontal cortex (lOFC) for punishment[5–8]. Second, reward and punishment learning can be selectively affected, for instance by dopaminergic manipulation and anterior insular lesion[9–12]. An important conclusion of these studies is that functional specificities relate to the learning domain (reward versus punishment) and not to the sign of prediction errors (positive versus negative). These two factors could be dissociated because reward learning also involves negative PE (when the expected reward is not delivered), while punishment learning also involves positive PE (when expected punishment is avoided). Thus, the suggestion is that some brain regions may signal reward PE (both positive and negative), informing whether or not a choice should be repeated, while other brain regions may signal punishment PE, informing whether or not a choice should be avoided.

However, a number of empirical studies have casted doubt on this anatomical separation between reward and punishment learning systems. Part of the confusion might come from the use of behavioral tasks that allow for a change of reference point, such that not winning becomes punishing and not losing becomes rewarding[13]. The issue is aggravated with decoding approaches that preclude access to the sign of PE signals, i.e., whether they increase or decrease with reward versus punishment[14]. Another reason for inconsistent findings might be related to the recording technique: fMRI instead of electrophysiology. Indeed, some electrophysiological studies in monkeys have recorded reward and punishment PE signals in adjacent brain regions[15,16]. In addition, single-unit recordings in monkeys have identified PE signals in other brain regions: not only small deep brain nuclei such as the ventral tegmental area[17,18] but also large cortical territories such as the dorsolateral prefrontal cortex (dlPFC, refs. [19,20]).

One issue with fMRI is that the temporal resolution makes it difficult to dissociate the two components of PE – observed and expected outcome value. The issue arises because the same region might reflect PE at both the times of option and outcome display. Thus, if option and outcome display are close in time, the hemodynamic signals reflecting positive and negative expected outcome value would cancel each other[21,22]. The issue can be solved by adequate jittering between cue and outcome events[23,24], but this has not been systematically used in human fMRI studies. In addition to recording techniques and related timing issues, discrepant results between human and monkey studies could also arise from differences in the paradigms[25], such as the amount of training or the particular reward and punishment used to condition choice behavior. Indeed, primary reinforcers used in monkeys like fruit juices and air puffs may not be exact reward and punishment equivalents, as are the monetary gains and losses used in humans.

With the aim of bridging across species and techniques, we investigate here PE signals in the human brain, using a time-resolved recording technique: intracerebral electroencephalography (iEEG). The iEEG signals were collected in patients implanted with electrodes meant to localize epileptic foci, while they performed an instrumental learning task. The same approach was used in one previous study that failed to identify any anatomical specificity in the neural responses to positive and negative outcomes[26]. To assess whether this lack of specificity was related to the recording technique or to the behavioral task, we used a task that properly dissociates between reward and punishment learning, as shown by previous fMRI, pharmacological, and lesion studies[6,11].

In this task (Fig. 1a), patients ($n = 20$) are required to choose between two cues to maximize monetary gains (during reward-learning) or minimize monetary losses (during punishment-learning). Reward and punishment PE can then be inferred from the history of tasks events, using a computational model. We first identifiy from the 1694 cortical recording sites a set of brain regions encoding PE, which include vmPFC, lOFC, aINS, and dlPFC. We then specify the dynamics of PE signals in both time and frequency domains, and compare between reward and punishment conditions. The main purpose of these analyses is to assess whether differences between brain regions relate to the sign (positive versus negative) or to the domain (reward versus punishment) of PE signals driving choice behavior.

## Results

iEEG data were collected from 20 patients with drug-resistant epilepsy (see demographical details in Supplementary Table 1 and Methods section) while they performed an instrumental learning task (Fig. 1a). Electrode implantation was performed according to routine clinical procedures, and all targeted brain areas for the presurgical evaluation were selected strictly according to clinical considerations with no reference to the current study.

Patients had to choose between two cues to either maximize monetary gains (for reward cues) or minimize monetary losses (for punishment cues). The pairs of cues associated to reward and punishment learning were intermingled within three to six sessions of 96 trials. In each pair, the two cues were associated to the two possible outcomes (0/1€ in the reward condition and 0/-1€ in the punishment condition) with reciprocal probabilities (0.75/ 0.25 and 0.25/0.75). Reward and punishment conditions were matched in difficulty, as the same probabilistic contingencies were to be learned. Patients were instructed to do their best to maximize the monetary gains and to minimize the monetary losses during the task. No further information was given regarding the exact task structure.

**Behavioral performance.** Patients were able to learn the correct response over the 24 trials of the learning session: they tended to choose the most rewarding cue in the reward condition and avoid the most punishing cue in the punishment condition (Fig. 1b). Average percentage of correct choices (Fig. 1c) in the reward and punishment conditions was significantly different from chance (50%) level (reward: $71.4 \pm 3.2\%$, $t_{19} = 6.69$, $p < 3 \times 10^{-6}$; punishment: $71.5 \pm 2.1\%$, $t_{19} = 10.02$, $p < 6 \times 10^{-9}$; difference: $t_{19} = -0.03$; $p = 0.98$; one-sample and paired-sample two-tailed Student's $t$ tests). Reaction times were significantly shorter in the reward than in the punishment condition (Fig. 1f; reward: $700 \pm 60$ ms; punishment: $1092 \pm 95$ ms; difference: $t_{19} = -7.02$, $p < 2 \times 10^{-6}$). Thus, patients learnt similarly from rewards and punishments, but took longer to choose between cues for punishment avoidance. This pattern of results replicates behavioral data previously obtained from healthy subjects[6,27].

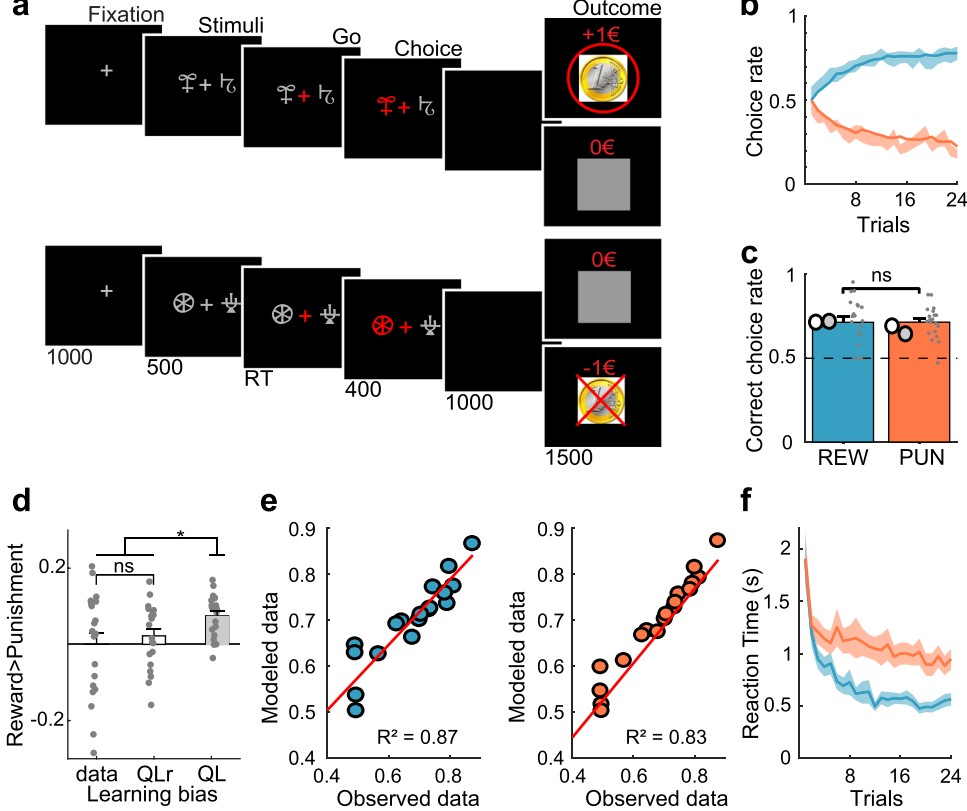

**Fig. 1 Behavioral task and results. a** Successive screenshots of a typical trial in the reward (top) and punishment (bottom) conditions. Patients had to select one abstract visual cue among the two presented on each side of a central visual fixation cross, and subsequently observed the outcome. Duration is given in milliseconds. **b** Average learning curves ($n = 20$ patients). Modeled behavioral choices (solid line) are superimposed on observed choices (shaded areas represent mean ± SEM across patients). Learning curves show rates of correct choice (75% chance of 1€ gain) in the reward condition (blue curves) and incorrect choice (75% chance of 1€ loss) in the punishment condition (red curves). **c** Average performance (correct choice rate, $n = 20$ patients). Modeled performance is indicated by white and gray disks (using Q-learning + repetition bias and basic Q-learning model, QLr and QL, respectively). Dots represent individual patients. **d** Difference between conditions (reward minus punishment correct choice rate) in observed and modeled data. Dots represent individual patients and error-bars represent mean ± SEM across patients ($n = 20$). **e** Inter-patient correlations between modeled and observed correct choice rate for reward (blue) and punishment (red) learning. Each circle represents one patient. Red line represents the linear regression across patients ($n = 20$). **f** Reaction time (RT) learning curves. Median RT are averaged across patients and the mean (±SEM) is plotted as function of trials separately for the reward (blue) and punishment (red) conditions. Black horizontal bars represent the outcome of two-sided statistical tests of difference, using paired Student's $t$ tests in **c** and **d**. ns means not significant and asterisk indicates significance in d (QL > QLr, $p = 0.0037$; QL > data, $p = 0.013$).

**Table 1 Model parameters and comparison criterion.**

| | Degrees of freedom (DF) | Bayesian information criterion (BIC) Mean ± SEM | Learning rate ($\alpha$) Mean ± SEM | Inverse temperature ($\beta$) Mean ± SEM | Repetition bias ($\theta$) Mean ± SEM |
|---|---|---|---|---|---|
| QL | 2 | 502 ± 31 | 0.27 ± 0.04 | 3.80 ± 0.48 | – |
| QLr | 3 | 430 ± 30 | 0.26 ± 0.04 | 3.19 ± 0.43 | 0.44 ± 0.06 |

**Computational modeling**. To generate trial-wise expected values and prediction errors, we fitted a Q-learning model (QL) to behavioral data. The QL model generates choice likelihood via a softmax function of cue values, which are updated at the time of outcome. Fitting the model means adjusting two parameters (learning rate and choice temperature) to maximize the likelihood of observed choices (see methods). Because this simple model left systematic errors in the residuals, we implemented another model (QLr) with a third parameter that increased the value of the cue chosen in the previous trial, thereby increasing the likelihood of repeating the same choice. We found that including a repetition bias in the softmax function better accounted for the data, as indicated by a significantly lower Bayesian Information Criterion (BIC) for QLr model ($t_{19} = 4.05$, $p < 0.001$; Table 1; one-sample

two-tailed Student's $t$ tests on the difference of BIC). On average, this QLr model accounts for a more symmetrical performance between reward and punishment learning (Fig. 1d), while the standard QL model would learn better in the reward condition, because reinforcement is more frequent than in the punishment condition (as patients approach the +1€ and avoid the −1€ outcome). With the QLr model, choices in reward and punishment conditions were captured equally well, with an explained variance across patients of 87 and 83% (Fig. 1e).

In the following analyses, iEEG activity was regressed against PE estimated for each participant at each trial in each condition from the QLr model fit. PE therefore represents our key independent variable – a hidden variable that in principle could have driven learning, since it is informed by individual choice behavior.

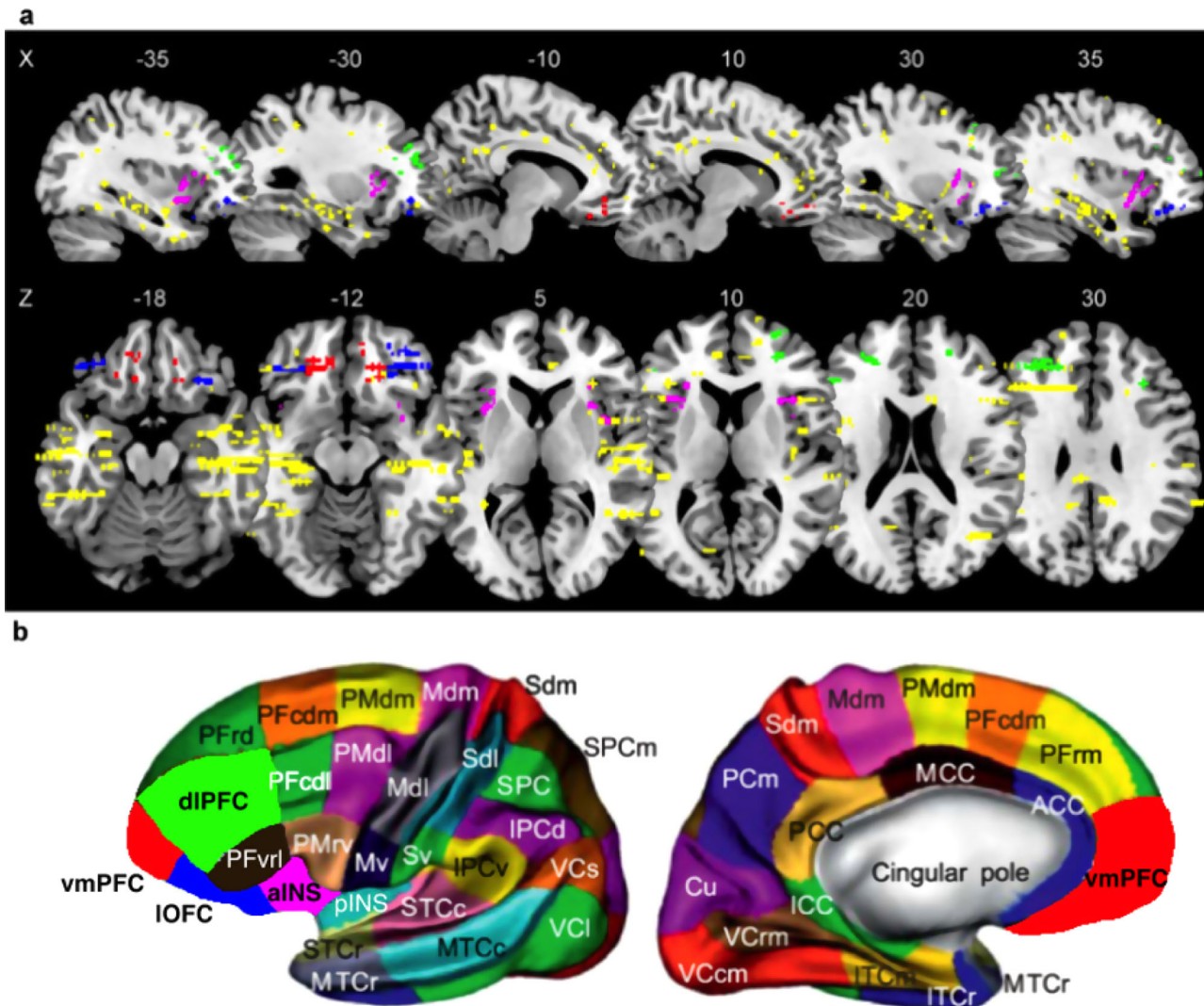

**Fig. 2 Anatomical location of intracerebral electrodes. a** Sagittal and axial slices of a brain template over which each dot represents one iEEG recording site ($n = 1694$). Color code indicates location within the four main regions of interest (red: vmPFC, $n = 54$; green: dlPFC, $n = 74$; blue: lOFC, $n = 70$; purple: aINS, $n = 83$). **b** MarsAtlas parcellation scheme represented on an inflated cortical surface.

**iEEG: localizing PE using broadband gamma activity**. To identify brain regions signaling PE, we first focused on broadband gamma activity (BGA, in the $50{-}150\,\text{Hz}$ range) because it is known to correlate with both spiking and fMRI activity[28–31]. BGA was extracted from each recording site and time point and regressed against PE (collapsed across reward and punishment conditions) which were generated by the QLr model across trials. The location of all iEEG recording sites ($n = 1694$ bipolar derivations) was labeled according to MarsAtlas parcellation[32], and to the atlas of Destrieux (Destrieux et al.[67]) for the hippocampus and the distinction between anterior and posterior insula (Fig. 2). In total, we could map 1473 recording sites into 39 brain parcels. In the following, we report statistical results related to PE signals tested across the recording sites located within a given parcel. Note that an inherent limitation to any iEEG study is that the number of recorded sites varies across parcels, which impacts the statistical power of analyses used to detect PE signals in different brain regions.

For each parcel, we first tested the significance of regression estimates (averaged over the 0.25–1 s time window following outcome onset) in a fixed-effect analysis (pooling sites across patients) and Bonferroni-corrected for multiple comparisons

across parcels. We also estimated the significance of PE signals at the site level, by using time-varying regression estimates and associated $p$-values, while FDR-correcting for multiple comparisons in the time domain (across 97 comparisons in the 0–1.5 s time window following outcome onset), in accordance with published methods[33]. We found eight parcels showing significant PE signals and displaying a proportion of significant contacts superior to 20% (Supplementary Table 2). This set of significant brain parcels included the aINS, vmPFC, dlPFC, lOFC, hippocampus, lateral, and caudal medial visual cortex (VCcm and VCl) and the medial inferior temporal cortex (ITcm). Given this result and the literature reviewed in the introduction, we focused on the anterior insula and prefrontal ROIs (vmPFC, lOFC, and dlPFC) in the hereafter analyses, while the same analyses performed in the other ROIs are presented as supplementary information.

**iEEG: PE signals across ROIs and frequency bands**. In each ROI (Fig. 3a), we explored whether activity in other frequency bands could also be related to PE (collapsed across reward and punishment conditions). We performed a time-frequency decomposition of the evoked response around outcome onset and

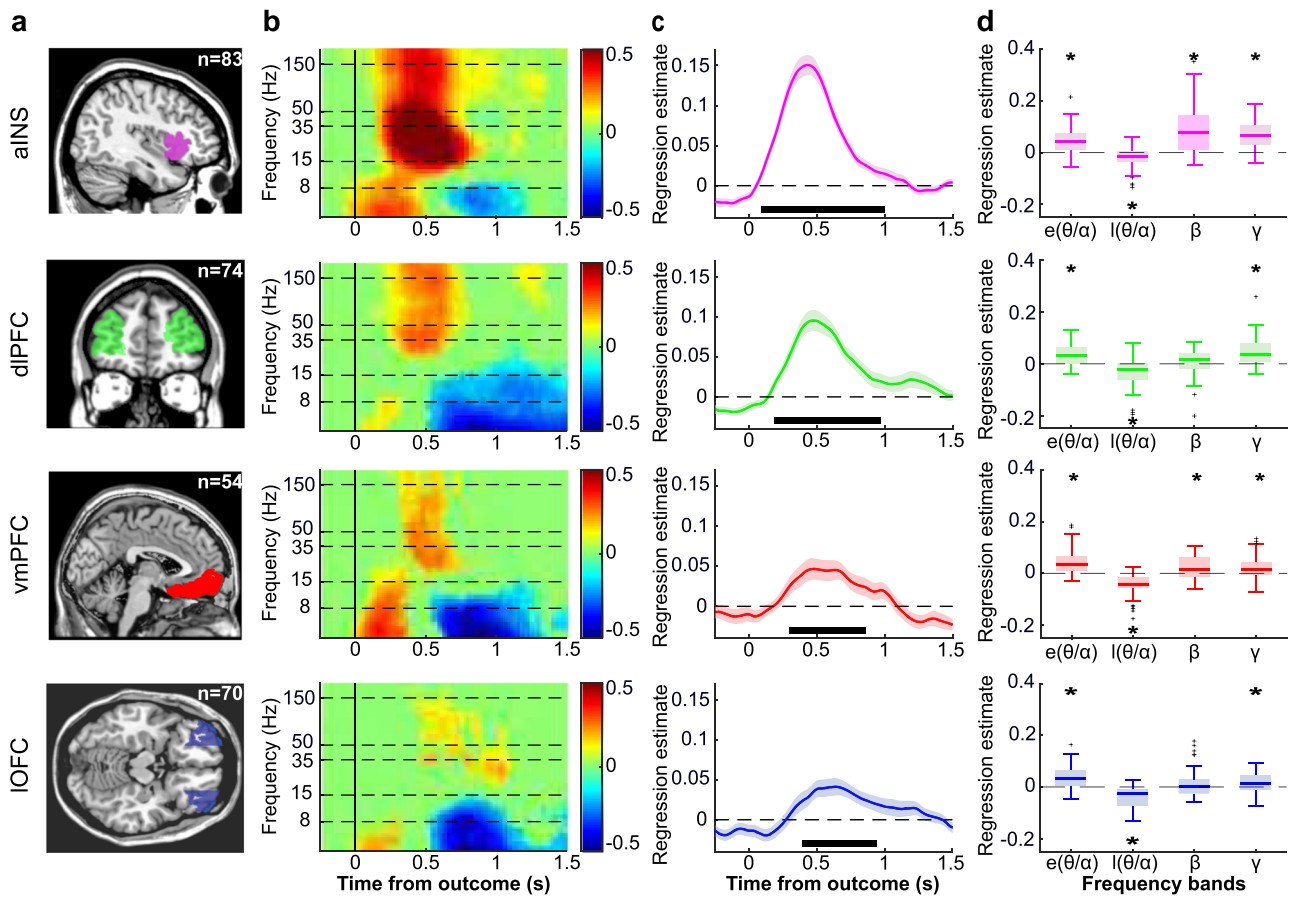

**Fig. 3 Investigation of PE signals across frequency bands. a** Anatomical localization of the aINS (purple), dlPFC (green), vmPFC (red), and lOFC (blue). All recording sites located in these parcels were included in the ROI analyses. The sample size used to derive statistics in panels **b**–**d** are displayed for each ROI (aINS: $n = 83$; dlPFC: $n = 74$; vmPFC: $n = 54$; lOFC: $n = 70$). **b**. Time-frequency decomposition of PE signals following outcome onset. Hotter colors indicate more positive regression estimates. Horizontal dashed lines indicate boundaries between frequency bands that are investigated in panels **c** and **d**. **c** Time course of regression estimates obtained from linear fit of BGA with PE modeled across reward and punishment conditions. Solid lines (filled areas) indicate mean (SEM) across recording sites. Horizontal bold black lines indicate significant clusters ($p_c < 1 \times 10^{-3}$; one-sample, two-sided Student's $t$ test after cluster-wise correction). **d** Regression estimates of power against PE, averaged over early (0–0.5 s) and late (0.5–1 s) post-stimulus windows for the lower-frequency bands ($\theta/\alpha$: 4–13 Hz) and over the 0.25–1 s window for higher frequency bands ($\beta$: 13–33 Hz and broadband $\gamma$: 50–150 Hz). Center lines, box limits, whiskers, and crosses of the box plots, respectively represent median, interquartile range, and outliers of the data distribution from the $n$ recording sites. Stars indicate significance (all $p$ values < 0.05) of regression estimates (one-sample, two-sided Student's $t$ test). Error bars correspond to inter-sites SEM and dots correspond to individual recording sites.

regressed each time and frequency point against PE generated by the QLr model across trials. This time-frequency analysis confirmed the presence of PE signals in BGA following outcome onset in all ROIs (Fig. 3b). Furthermore, PE was also positively associated with beta-band (13–33 Hz) power in the aINS and vmPFC. In the theta/alpha bands (4–8 and 8–13 Hz), there was an initial positive association (during the first 500 ms after outcome onset), which was followed by a negative association (from 500 to 1000 ms after outcome onset) in all four ROIs. Thus, the time-frequency analysis pointed to three other frequency bands in which power could be associated to PE.

To confirm this observation, we regressed trial-wise power against PE, in the four ROIs and four frequency bands, for each time point between −0.2 and 1.5 s around outcome onset. In the broadband gamma (Fig. 3c), we found a significant cluster-corrected association with PE in the 0.09–1.00 s window for the aINS ($\beta_{PE} = 0.08 \pm 0.007$, sum($t_{(82)}$) = 462.2, $p_c < 1 \times 10^{-3}$, one-sample, two-tailed Student's $t$ test after cluster-wise correction), 0.19–0.97 s for the dlPFC ($\beta_{PE} = 0.06 \pm 0.008$, sum($t_{(73)}$) = 273.5, $p_c < 1 \times 10^{-3}$), 0.30–0.86 s for the vmPFC ($\beta_{PE} = 0.03 \pm 0.008$, sum($t_{(53)}$) = 115.3, $p_c < 1 \times 10^{-3}$) and 0.39–0.94 s for the lOFC

($\beta_{PE} = 0.03 \pm 0.006$, sum($t_{(69)}$) = 116.1, $p_c < 1 \times 10^{-3}$; Fig. 3c). We next focused on a 0.25–1 s post-outcome time window for subsequent analyses, as it plausibly corresponds to the computation of PE. To further quantify statistically how the information about PE was distributed across frequencies, we averaged regression estimates over the 0.25–1 s time window for the broadband gamma and beta bands, and over two separate time windows to distinguish the early (0–0.5 s) and late (0.5–1 s) components of theta-alpha band activity (Fig. 3d). As expected, we found significant PE correlates in BGA in the four ROIs (all $p < 0.05$). Furthermore, beta-band activity was also positively associated with PE in two ROIs (aINS: $\beta_{PE} = 0.11 \pm 0.012$; $t_{(82)}$ = 9.17; $p < 1 \times 10^{-13}$; vmPFC: $\beta_{PE} = 0.03 \pm 0.008$; $t_{(53)} = 3.34$; $p = 0.0015$, one-sample two-tailed Student's $t$ test). Finally, regarding the theta/alpha band, regression estimates were significantly above (below) zero in the early (late) time window in all ROIs (all $p$ values < 0.05).

To compare the contribution of activities in the different frequency bands to PE signaling across the four ROIs, we included them as separate regressors in general linear models meant to explain PE. The general aim of this analysis was to test

whether lower-frequency bands were adding any information about PE (compared to BGA alone). We thus compared GLMs including only BGA to all possible GLMs containing broadband gamma plus any combination of low-frequency activities. Bayesian model selection (see Methods) designated the broadband-gamma-only GLM as providing the best account of PE (Ef = 0.997, Xp = 1). Thus, even if low-frequency activity was significantly related to PE, it carried redundant information relative to that extracted from BGA.

**iEEG: comparison between reward and punishment PE**. In the following analyses, we focused on BGA and tested whether prediction errors estimated in the reward (RPE = R-Qr) and punishment (PPE = P-Qp) conditions could be dissociated between the four ROIs previously identified (aINS, dlPFC, vmPFC, and lOFC). We computed the time course of regression estimates separately for reward and punishment PE. We observed an increase of the regression estimate at the time of outcome display, which differed between conditions and ROIs (Fig. 4a). In aINS and dlPFC, regression estimates were significantly higher for punishment than for reward PE, in the 0.23–0.70 s window for the aINS ($\beta_{RPE} - \beta_{PPE} = -0.06 \pm 0.02$, sum($t_{(82)}$) = −97.01, $p_c < 1 \times 10^{-3}$, one-sample two-tailed Student's $t$ test after cluster-wise correction) and in the 0.25–1.5 s for the dlPFC ($\beta_{RPE} - \beta_{PPE} = -0.06 \pm 0.01$, sum($t_{(73)}$) = −368.6, $p_c < 1 \times 10^{-3}$). An opposite pattern emerged in the vmPFC and lOFC: regression estimates were significantly higher for reward PE than for punishment PE, in the 0.48–1.02 s for the vmPFC ($\beta_{RPE} - \beta_{PPE} = 0.06 \pm 0.01$, sum ($t_{(53)}$) = 116, $p_c < 1 \times 10^{-3}$) and in the 0.72–1.45 s for the lOFC ($\beta_{RPE} - \beta_{PPE} = 0.04 \pm 0.01$, sum($t_{(69)}$) = 138.7, $p_c < 1 \times 10^{-3}$).

To confirm this dissociation between reward PE and punishment PE signaling regions, we performed a number of control analyses, at different levels from iEEG data preprocessing to model-based regressions. First, the functional dissociation was unchanged when using an alternative procedure for extracting BGA (see Methods section and Supplementary Fig. 1) or after removing recording sites with pathological activity and trials with artifacts (see Methods section and Supplementary Fig. 2).

Second, we examined whether the functional dissociation could arise from a differential involvement in the different leaning phases, as patients typically get more punishments at the beginning and more reward at the end of a learning session. Also, during the course of leaning, patients could figure out the mapping between the different pairs of cues and reward versus punishment domains and hence reframe their expectations. We checked that the difference between RPE and PPE signals observed across the four regions of interest was still significant even after controlling for trial index within learning sessions (Supplementary Fig. 3). We also checked that the magnitude of PE signals was constant throughout the learning session. For this we simply used the contrast between possible outcomes (reward versus no reward and punishment versus no punishment), in a model-free analysis. In the RL framework, this contrast should be stable because it does not depend on what is learned (i.e., on expectations). Indeed, this contrast of BGA activity following zero and non-zero outcomes was of similar magnitude in early and late trials of learning sessions (Supplementary Fig. 4). This was observed in both reward PE (vmPFC and lOFC) and punishment PE (aIns and dlPFC) signaling ROIs. Thus, PE were reliably signaled from the beginning to the end of learning, with a stable difference between ROIs more sensitive to reward versus punishment outcomes.

Third, we examined whether the differences between ROIs signaling reward and punishment PE could arise from differences in the proportion of recording sites sensitive to reward versus punishment (Supplementary Fig. 5). In the aIns and dlPFC, recording sites were more likely to represent punishment prediction errors whereas in the vmPFC and lOFC, recording sites were more likely to represent reward prediction errors.

**iEEG: breaking PE into outcome and expectation signals**. We next decomposed PE signals into outcome and expected value, for both reward (R and Qr) and punishment (P and Qp), to test whether the two components were reflected in BGA, at the time of outcome display. We observed a consistent pattern across ROIs: while the outcome component was positively correlated with BGA, the expectation component was negatively correlated with BGA (Fig. 4b). To further quantify this pattern in each ROI, we tested regression estimates (averaged over a 0.25–1-s post-outcome time window) of both outcome and expectation for both reward and punishment PE (Fig. 4c). We found that both components of PE were significantly expressed in the vmPFC and lOFC BGA following reward outcomes and in the aIns and dlPFC BGA following punishment outcomes (all $p$ values < 0.05). Because baseline correction might affect the decrease in BGA observed with expected reward or punishment, we checked that the results were robust (Supplementary Fig. 6) to a change in the time window used to extract baseline activity (500 ms pre-fixation epoch instead of 6 s around outcome onset).

We next examined whether expectation signals could be observed during the delay period preceding outcome delivery, by averaging regression estimates over a −1.2 to −0.2 time window before outcome onset. We found (Supplementary Fig. 7) that expectations tended to be positively related to BGA in all regions but regression estimates were significantly above zero for expected punishment only (Qp in aIns and dlPFC), and not for expected reward (Qr in vmPFC and lOFC).

Finally, we explored the effects of domains (reward vs. punishment PEs) in the four other brain regions associated with PE (Supplementary Fig. 8). BGA in the medial Inferior Temporal Cortex (mITC) was associated with outcomes (reward and punishments), but not with expected value, so this region would not qualify as signaling PE. The three other ROIs (hippocampus (HPC), lateral visual cortex (lVC), and caudal medial visual cortex: (cmVC)), showed a dissociation in time, with a short punishment PE and prolonged reward PE. When averaging the signal over the 0.25–1-s post-outcome time window, there was no significant difference between reward and punishment PE in these regions (all $p$ values < 0.05). There was therefore no strong evidence for these regions to be associated with either reward or punishment learning.

## Discussion

Here, we compared the neural correlates of reward and punishment PE during instrumental learning. We identified a set of brain regions signaling PE in different frequency bands, the most informative being BGA. All regions signaled outcomes with increased BGA and expectations with decreased BGA. However, there was a partial dissociation: the vmPFC and lOFC emitted stronger signals for reward PE, whereas the aINS and dlPFC emitted stronger signals for punishment PE. This anatomical divide relates to the learning domain (reward versus punishment) and not to the sign of PE, since in all cases outcomes (whether positive or negative) were signaled with increased BGA. In the following, we successively discuss the specification of PE signals in terms of anatomical location and frequency band, and then the dissociation between reward and punishment PE.

When regressing BGA against PE modeled across learning conditions, we identified significant correlates in a number of brain regions. Among the significant ROIs, some (e.g., the vmPFC

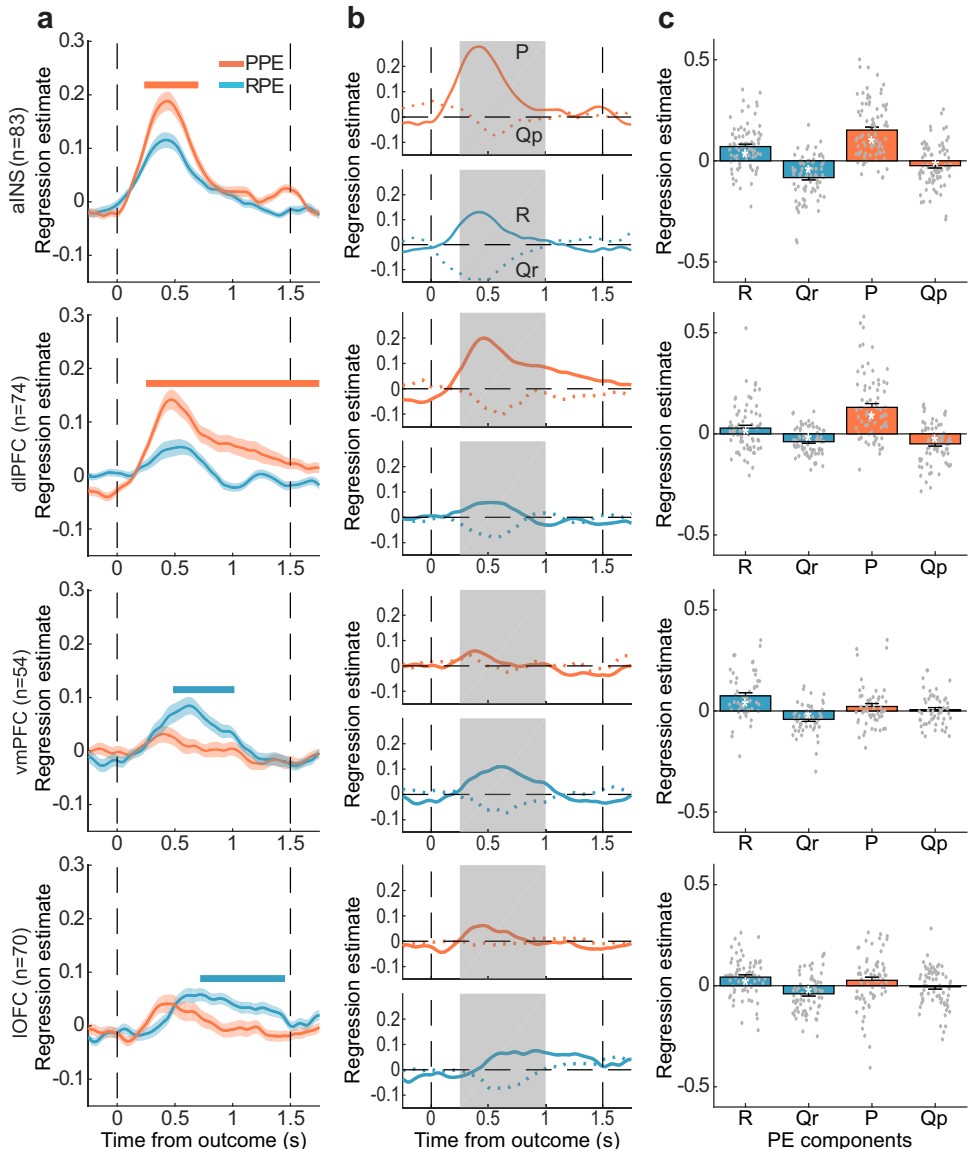

**Fig. 4 Dissociation of reward PE (R-Qr) and punishment PE (P-Qp) signals. a** Time course of regression estimates obtained from linear fit of BGA with PE modeled separately for the reward (blue) and punishment (red) conditions (PPE punishment prediction error, RPE reward prediction error). Horizontal bold lines indicate significant difference between conditions (blue: RPE > PPE; red: PPE > RPE; pc < 0.05). Shaded areas represent inter-sites SEM. **b** Time course of regression estimates obtained from a linear model including both outcome (solid lines) and expected value (dotted lines) components for both reward (R and Qr) and punishment (P and Qp) PE. **c** Regression estimates averaged over the 0.25–1 s time window (represented as shaded gray areas in panels **b**). Stars indicate significance (*$p < 0.05$, one-sample, two-tailed Student's $t$ test). Error-bars correspond to inter-sites SEM and dots correspond to individual recording sites. The sample size ($n$) used to derive statistics in all panels was: aINS: $n = 83$ sites; dlPFC: $n = 74$; vmPFC: $n = 54$; lOFC: $n = 70$.

and aINS) were classic regions related to prediction errors in meta-analyses of human fMRI studies[34–36], whereas others (e.g., the dlPFC and lOFC) were regions where single-neuron firing activity in non-human primates was shown to correlate with prediction error[19,20,37]. Our study thus fills a gap across species and techniques, confirming that intracerebral BGA is a relevant neurophysiological signal related to both hemodynamic and spiking activity, as previously suggested[28–31].

Yet it raises the question of why fMRI studies, including those using the same task as here[6], often failed to detect PE correlates in regions such as the dlPFC and lOFC. One possible explanation is the more stringent correction for multiple comparisons across voxels in fMRI studies, compared to the correction across ROIs applied here, or the absence of correction in most animal studies that typically investigate a single brain region. Another

explanation would be that regions such as the dlPFC are more heterogenous across individuals, such that the group-level random-effect analyses typically conducted in fMRI studies might be less sensitive than the fixed-effect analyses performed here or in animal studies. Conversely, some key regions consistently found to signal PE in fMRI studies (e.g., the ventral striatum) are absent in our results, for the simple reason that they were not sampled by the electrodes implanted for clinical purposes. Even if our results provide some insights about the location of PE signals, they cannot be taken as arising from a fair whole-brain analysis, since some regions were more frequently sampled than others, biasing the statistical power and hence the sensitivity of PE detection.

In all the investigated ROIs, we also found significant links with activity in lower-frequency bands. Time-frequency decomposition of PE correlates yielded remarkably similar patterns in the

different ROIs, with an increase in the beta to high-gamma band, and an increase followed by a decrease in the theta to alpha band. The late decrease may reflect the anti-correlation between theta-band activity and the other signals (broadband gamma, hemo-dynamic, and spiking activity) that was documented in previous studies[29,30,38]. However, the early increase is more surprising and suggests that PE are initially signaled in low-frequency activity, before reaching BGA. Yet when the different frequency bands were put in competition for predicting PE across trials, low-frequency activity proved to be redundant, with respect to the information already contained in BGA. This result is in line with our results regarding subjective valuation for decision making[39]: all the information available in neural activity could be found in BGA, even if activity in lower-frequency bands also showed significant value signals.

The timing of PE signals, peaking ~0.5 s after outcome onset, was roughly compatible with that observed in the hemodynamic response, which is typically delayed by 5–6 s. The (positive) correlation with the outcome and the (negative) correlation with the expectation were simultaneously observed after outcome display. Interestingly, expectations tended to be positively associated with BGA, the sign of the correlation being reverted when the outcome was delivered. These observations, made possible here by the high temporal resolution of iEEG, are rarely reported in fMRI studies[40]. One reason is that the hemodynamic response, because of its low temporal resolution, may confound positive expectation at cue onset and negative expectation at outcome onset, unless the two events are separated by a long delay (as in, e.g. refs. [23,24]). Our double observation corroborates a previous study showing that the differential response to positive and negative feedbacks, recorded with intracranial electrodes, is modulated by reward expectation[26]. As the response to outcome can be viewed as an indicator of valence, and the modulation by expectation as an effect of surprise, it shows that valence and surprise can be represented in the same brain region, in accordance with the very notion of prediction error signal.

Although all our ROIs exhibited a similar pattern of response, they differ in the strength of reward versus punishment PE signals. This anatomical dissociation between learning systems in the brain may appear at variance with previous studies reporting that rewards and punishments are ubiquitously represented all over the brain[14,26]. However, these non-specific results were observed during tasks in which the outcome is either reward or punishment. Thus, it is understandable that both reward and punishment regions were mobilized by the outcome in these previous studies, since being rewarded is not being punished and vice-versa. Here, PE signals were defined by the comparison between reward or punishment outcomes and their omission, not with each other, which enabled a dissociation. Besides, the previous conclusion was based on the finding that the information about reward versus punishment outcomes could be recovered from many brain regions. Had we applied the same decoding analysis here, we would have reached the same conclusion: information about reward versus punishment PE could be recovered in all our ROIs, precisely because their response depended on outcome valence. In other words, the contrast between reward and punishment PE would be significant in all ROIs, precisely because they are differentially sensitive to the two learning domains. The dissociation between regions signaling reward and punishment PE may also seem at odds with single-unit recordings showing reward and punishment PE signals can be found in neighboring neurons[15,16]. Yet it should be emphasized that the dissociation observed here was only partial, compatible with the possibility that some regions contained more reward-sensitive neurons and others more punishment-sensitive neurons, even if both types can be found in all regions.

An important conclusion of our analyses is that the dissociation was made between reward and punishment PE, not between positive and negative PE. Indeed, some learning models assume that positive and negative PE are processed differently, yielding different learning rates (e.g. refs. [41,42]). A strict dissociation between positive and negative PE (across valence) would imply that regions signaling reward PE with increased activity would signal punishment PE with decreased activity, and vice-versa. This would induce an ambiguity for the rest of the brain, as an omitted reward would be coded similarly to an inflicted punishment, and an avoided punishment similarly to an obtained reward. This is not the pattern that we observed: on the contrary, both reward and punishment PE were positively correlated to BGA in all regions (at least numerically, if not significantly). Yet reward and punishment PE could be distinguished by a downstream region, from the relative activity of regions more sensitive to reward and those more sensitive to punishment. Thus, rather than the sign of PE, the dissociation depended on their domain, i.e., on whether the PE should reinforce the repetition or the avoidance of last choice.

Within the reward-sensitive regions, the vmPFC was expected, given the number of fMRI studies reporting a link between vmPFC and reward outcome, including those using the same task as here (Pessiglione et al.[6] reanalyzed in Palminteri et al.[11]) and meta-analyses[34–36,43]. The expression of reward PE in the vmPFC might relate to its position as a main efferent output of midbrain dopamine neurons, following the meso-cortical pathway[44]. Indeed, manipulation of dopaminergic transmission was found to interfere with reward learning, specifically[9,10,22], an effect that was captured by reward sensitivity in a computational model of learning in this task[6]. The observation of reward PE signals in the lOFC was less expected, because it is generally not reported in meta-analyses of human fMRI studies and because several electrophysiology studies in animals suggested that, even if orbito-frontal cortex neurons respond to reward outcomes, they might not encode prediction errors[45,46]. However, the similarity between lOFC and vmPFC reward PE signals is consistent with previous iEEG studies showing similar representation of subjective value and reward outcome in the two regions BGA[39,47]. Yet the lOFC and vmPFC reward PE signals may serve different functions, as was suggested by lesion studies in both human and non-human primates showing that the lOFC (but not the vmPFC) is critical for solving the credit assignment problem[48,49].

Within the punishment-sensitive regions, the aINS was expected, as it was associated with punishment PE in our fMRI study using the same task[6] and because it is systematically cited in meta-analyses of fMRI studies searching for neural correlates of punishment outcomes[35,36,40]. Surprisingly, the link between aINS activity and punishment PE has seldom been explored in non-human primates. This exploration was made possible here by the development of oblique positioning techniques employed to implant electrodes, which result in a large spatial sampling of the insular cortex[50]. This is important because other iEEG approaches, such as subdural recordings (Ecog), could not explore the role of the insular cortex in instrumental learning[26]. The present result echoes a previous finding, using the same technique, that aINS BGA signals mistakes in a stop-signal task[51]. By comparison, punishment PE signals in the dlPFC were less expected, since they were not observed in fMRI results using the same task, even if it is not uncommon to observe dlPFC activation following punishment outcomes[35,36,40]. The dissociation observed here at the cortical level between reward versus punishment PE signals might be related to afferences from different neuromodulatory systems, such as dopaminergic versus serotonergic pathways, which have been suggested to play opponent roles in approach versus avoidance behaviors[3,52].

Reward PE signals were also observed in both aINS and dlPFC regions, albeit with a lesser sensitivity. This may be interpreted as an effect of saliency rather than PE[22,53], as punishments were less frequent in the task than rewards (because patients learned to avoid the former and obtain the latter). However, pure saliency coding would not explain the responses to punishments observed in the aINS during Pavlovian learning tasks where high punishments were controlled to be more frequent than low punishments (e.g. Seymour et al.[54]) or in gambling tasks where punishment and reward outcomes were matched (e.g. Petrovic et al.[55]). Also, saliency coding would not predict the consequence of aINS damage, which was found to specifically impair punishment learning in this task, an effect that was captured by a specific diminution of the sensitivity to punishment outcome in a computational model[11]. Yet it remains that reward and punishment learning are not exact symmetrical processes, since positive reward PE favors repetition of the same choice, whereas positive punishment PE pushes to the alternative choice, hence involving an additional switching process. This switching process might explain the longer choice RT observed in the punishment condition. The switch might relate to the prolonged implication of the dlPFC following punishment PE, in keeping with the established role of this region in cognitive control[56–58]. The implication of the aINS might be more related to the aversiveness of punishment PE, in line with the role attributed to this region in pain, interoception, and negative feelings[59–61].

The data have been collected in patients being treated for epilepsy. It is unlikely that disease or treatment affected learning performance because the behavior of our patients was comparable to that of healthy young participants performing a similar task (e.g. Palminteri et al.[27]). We made the assumption that epileptic activity did not distort the brain signals linked to PEs and interpreted the data as if they were collected in healthy individuals. Furthermore, epileptic artifacts are unlikely to covary with the computational variable (PE) against which brain activity was regressed. We formally verified that removing artifacts did not affect the results about PE signals observed in BGA. We suggest that artifact removal may be unnecessary in model-based analyses of iEEG activity, and that results may actually be more robust if based on the entire dataset without ad-hoc selection based on visual inspection.

In summary, we used human intracerebral BGA to test the a priori theoretical principle that reward and punishment PE could be processed by the same brain machinery (one being the negative of the other). On the contrary, we found that both reward and punishment PE were positively correlated to BGA in all brain regions. Yet some regions amplified reward PE signals, and others punishment PE signals. Thus, the dissociation between reward and punishment brain systems is not about the sign of the correlation with PE, but about the valence domain of outcomes (better or worse than nothing). These appetitive and aversive domains correspond to different behaviors that must be learned: more or less approach for reward PE and more or less avoidance for punishment PE. Further research is needed to disentangle the roles of the different reward and punishment regions in these learning processes.

## Methods

**Patients**. Intracerebral recordings were obtained from 20 patients (33.5 ± 12.4 years old, 10 females, see demographical details in Supplementary Table 1) suffering from pharmaco-resistant focal epilepsy and undergoing presurgical evaluation. They were investigated in two epilepsy departments (Grenoble and Lyon). To localize epileptic foci that could not be identified through noninvasive methods, neural activity was monitored in lateral, intermediate, and medial wall structures in these patients using stereotactically implanted multilead electrodes (stereotactic intracerebral electroencephalography, iEEG). All patients gave written informed consent and the study received approval from the ethics committee (CPP 09-

CHUG-12, study 0907) and from a competent authority (ANSM no: 2009-A00239-48).

**iEEG data acquisition and preprocessing**. Patients underwent intracerebral recordings by means of stereotactically implanted semirigid, multilead depth electrodes (sEEG). In total, 5–17 electrodes were implanted in each patient. Electrodes had a diameter of 0.8 mm and, depending on the target structure, contained 8–18 contact leads 2-mm-wide and 1.5-mm-apart (Dixi, Besançon, France). Anatomical localizations of iEEG contacts were determined on the basis of post-implant computed tomography scans or postimplant MRI scans coregistered with preimplant scans[62]. Electrode implantation was performed according to routine clinical procedures, and all target structures for the presurgical evaluation were selected strictly according to clinical considerations with no reference to the current study.

Neuronal recordings were conducted using an audio–video-EEG monitoring system (Micromed, Treviso, Italy), which allowed simultaneous recording of 128–256 depth-EEG channels sampled at 256 Hz (1 patient, note that this patient was removed from analyses based on broadband gamma activity which could not be computed given the low sampling rate), 512 Hz (6 patients), or 1024 Hz (12 patients) [0.1–200 Hz bandwidth]. One of the contacts located in the white matter was used as a reference. Each electrode trace was subsequently re-referenced with respect to its direct neighbor (bipolar derivations with a spatial resolution of 3.5 mm) to achieve high local specificity by canceling out effects of distant sources that spread equally to both adjacent sites through volume conduction[63].

In order to take advantage of our large sample set, all recording sites with an anatomical label were included in the analyses (i.e., without any exclusion of sites with artifacts or pathological activity). Nevertheless, to check whether electrodes with artifacts or pathological activity could have biased the results, we applied a semi-automatic pipeline: first, bad channels detection was conducted with a machine learning approach[64], secondly, epileptic spikes were detected automatically with Delphos – Detector of ElectroPhysiological Oscillations and Spikes –[65] and all data were then finally visually inspected to check their quality. We also excluded recording sites that were part of the epileptogenic zone by identifying with the neurologists (PK and SR) all recording sites involved at seizure onset and/or sites that were located within the cortical resection (if any) performed after the sEEG; furthermore, trials during which iEEG activity was higher or lower than four times the standard deviation of the average signal were excluded. The results were not affected by this procedure.

**Behavioral task**. Patients performed a probabilistic instrumental learning task adapted from previous studies[6,11]. Patients were provided with written instructions, which were reformulated orally if necessary, stating that their aim in the task was to maximize their financial payoff and that to do so, they had to consider reward-seeking and punishment-avoidance as equally important (Fig. 1). Patients performed short training sessions to familiarize with the timing of events and with response buttons. Training procedure comprised a very short session, with only two pairs of cues presented on 16 trials, followed by 2–3 short sessions of 5 min such that all patients reached a threshold of 70% correct choices during both the reward and punishment conditions. During iEEG recordings, patients performed three to six test sessions on a single testing occurrence (with short breaks between sessions). Each session was an independent task containing four new pairs of cues to be learned. Cues were abstract visual stimuli taken from the Agathodaimon alphabet. Each pair of cues was presented 24 times for a total of 96 trials. The four cue pairs were divided in two conditions (2 pairs of reward and 2 pairs of punishment cues), associated with different pairs of outcomes (winning 1€ versus nothing or losing 1€ versus nothing). The reward and punishment conditions were intermingled in a learning session and the two cues of a pair were always presented together. Within each pair, the two cues were associated to the two possible outcomes with reciprocal probabilities (0.75/0.25 and 0.25/0.75). On each trial, one pair was randomly presented and the two cues were displayed on the left and right of a central fixation cross, their relative position being counterbalanced across trials. The subject was required to choose the left or right cue by using their left or right index to press the corresponding button on a joystick (Logitech Dual Action). Since the position on screen was counterbalanced, response (left versus right) and value (good versus bad cue) were orthogonal. The chosen cue was colored in red for 250 ms and then the outcome was displayed on the screen after 1000 ms. In order to win money, patients had to learn by trial and error the cue–outcome associations, so as to choose the most rewarding cue in the reward condition and the less punishing cue in the punishment condition. Visual stimuli were delivered on a 19 inch TFT monitor with a refresh rate of 60 Hz, controlled by a PC with Presentation 16.5 (Neurobehavioral Systems, Albany, CA).

**Behavioral analysis**. Percentage of correct choice (i.e., selection of the most rewarding or the less punishing cue) and reaction time (between cue onset and choice) were used as dependent variables. Statistical comparisons between reward and punishment learning were assessed using two-tailed paired $t$-tests. All statistical analyses were performed with MATLAB Statistical Toolbox (MATLAB R2017a, The MathWorks, Inc., USA).

**Computational modeling.** A standard Q-learning algorithm (QL) was used to model choice behavior. For each pair of cues, A and B, the model estimates the expected value of choosing A (Qa) or B (Qb), according to previous choices and outcomes. The initial expected values of all cues were set at 0, which corresponded to the average of all possible outcome values. After each trial ($t$), the expected value of the chosen stimuli (say A) was updated according to the rule:

$$Qa_{t+1} = Qa_t + \alpha * \delta_t \tag{1}$$

The outcome prediction error, $\delta(t)$, is the difference between obtained and expected outcome values:

$$\delta_t = R_t + Qa_t \tag{2}$$

with $R(t)$ the reinforcement value among −1€, 0€, and +1€. Using the expected values associated with the two possible cues, the probability (or likelihood) of each choice was estimated using the softmax rule:

$$Pa_t = \frac{e^{Qa_t/\beta}}{e^{Qa_t/\beta} + e^{Qb_t/\beta}} \tag{3}$$

The constant parameters $\alpha$ and $\beta$ are the learning rate and choice temperature, respectively. A second Q-Learning model (QLr) was implemented to account for the tendency to repeat the choice made on the preceding trial, irrespective of the outcome. A constant ($\theta$) was added in the softmax function to the expected value of the option chosen on the previous trial presented the same cues. For example, if a subject chose option A on trial $t$:

$$Pa_{t+1} = \frac{e^{Qa_t+\theta/\beta}}{e^{Qa_t+\theta/\beta} + e^{Qb_t/\beta}} \tag{4}$$

We optimized model parameters by minimizing the negative log likelihood ($LL_{max}$) of choice data using MATLAB fmincon function, initialized at multiple starting points of the parameter space, as previously described (Palminteri et al.[27]). Bayesian information criterion (BIC) was computed for each subject and model:

$$BIC = \log(ntrials) \times (n\ degrees\ of\ freedom) + 2 \times LL_{max} \tag{5}$$

Outcome prediction errors (estimated with the QLr model) for each patient and trial were then Z-scored and used as statistical regressors for iEEG data analysis

**Electrophysiological analyses.** Collected iEEG signals were analyzed using Fieldtrip[66] and homemade MATLAB codes. Anatomical labeling of bipolar derivation between adjacent contact-pairs was performed with IntrAnat software[62]. The 3D T1 pre-implantation MRI gray/white matter was segmented and spatially normalized to obtain a series of cortical parcels using MarsAtlas[32] and the Destrieux atlas[67]. 3D models of electrodes were then positioned on post-implantation images (MRI or CT). Each recording site (i.e., each bipolar derivation) was thus labeled according to its position in a parcellation scheme in the patients' native space. Thus, the analyzed dataset only included electrodes identified to be in the gray-matter.

*Regions of interest definition.* The vmPFC ROI (54 sites) was defined as the ventromedial PFC plus the fronto-medial part of orbitofrontal cortex bilaterally (MarsAtlas labels: PFCvm plus mesial part of OFCv and OFCvm). The lOFC ROI ($n = 70$ sites) was defined as the bilateral central and lateral parts of the orbito-frontal cortex (MarsAtlas labels: OFCvl plus lateral parts of OFCv). The dlPFC ROI ($n = 74$ sites) was defined as the inferior and superior bilateral dorsal prefrontal cortex (MarsAtlas labels: PFrdli and PFrdls). The aINS ROI ($n = 83$ sites) was defined as the bilateral anterior part of the insula (Destrieux atlas labels: Short insular gyri, anterior circular insular sulcus and anterior portion of the superior circular insular sulcus).

*Computation of single-trial broadband gamma envelopes.* Broadband gamma activity (BGA) was extracted with the Hilbert transform of iEEG signals using custom MATLAB scripts as follows. iEEG signals were first bandpass filtered in 10 successive 10-Hz-wide frequency bands (e.g., 10 bands, beginning with 50–60 Hz up to 140–150 Hz). For each bandpass filtered signal, we computed the envelope using standard Hilbert transform. The obtained envelope had a time resolution of 15.625 ms (64 Hz). Again, for each band, this envelope signal (i.e., time-varying amplitude) was divided by its mean across the entire recording session and multiplied by 100 for normalization purposes. Finally, the envelope signals computed for each consecutive frequency bands (e.g., 10 bands of 10 Hz intervals between 50 and 150 Hz) were averaged together, to provide one single time-series (the BGA) across the entire session, expressed as percentage of the mean. This procedure was used to counteract a bias toward the lower frequencies of the frequency interval induced by the 1/f drop-off in amplitude. Finally, these time-series were smoothed with a 250 ms sliding window to increase statistical power for inter-trial and inter-individual analyses of BGA dynamics. This procedure was previously shown to maximize the signal/noise ratio to detect task-related modulation of BGA by effectively smoothing the signal across frequencies and time to detect BGA modulations that exhibit across trials and across sites variability in terms of precise timing and frequency signature[28,39,51,68–73]. In addition, we also checked whether estimating BGA using a single bandpass filter across the whole

frequency band of interest[47] instead of filtering across consecutive frequency bands yielded to a similar pattern of result (Supplementary Fig. 1). For each trial, baseline correction (Z-score) was applied using the average and standard deviation computed over a 6-s epoch centered around outcome onset (−3 to 3 s). We also verified that using a 500-ms fixation epoch preceding the outcome to z-score each trial yielded similar results.

*Computation of envelopes in lower frequencies.* The envelopes of theta, alpha and beta bands were extracted in a similar manner as the broadband gamma frequency except that steps were 1 Hz for $\theta$ and $\alpha$ and 5 Hz for $\beta$. The ranges corresponding to the different frequency bands were as follows: broadband gamma was defined as 50–150 Hz, beta as 13–33 Hz, alpha as 8–13 Hz, and theta as 4–8 Hz. For each trial, baseline correction (Z-score) was applied using the average and standard deviation computed over a 6-s epoch centered around outcome onset (−3 to 3 s).

*Time-frequency decomposition.* Time-frequency analyses were performed with the FieldTrip toolbox for MATLAB. A multitapered time-frequency transform allowed the estimation of spectral powers (Slepian tapers; lower-frequency range: 4–32 Hz, 6 cycles and 3 tapers per window; higher frequency range: 32–200 Hz, fixed time-windows of 240 ms, 4–31 tapers per window). This approach uses a steady number of cycles across frequencies up to 32 Hz (time window durations therefore decrease as frequency increases) whereas for frequencies above 32 Hz, the time window duration is fixed with an increasing number of tapers to increase the precision of power estimation by increasing smoothing at higher frequencies.

*General linear models.* Frequency envelopes of each recording site were epoched on each trial and time locked to the outcome onset (−3000 to 1500 ms). Each time series was regressed against the variables of interest to obtain a regression estimate per time point and recording site. In all GLMs, normalized power (Y) was regressed across trials against prediction error signal PE (normalized within patients) at every time point:

$$Y = \alpha + \beta \times PE \tag{6}$$

with $\beta$ corresponding to the regression estimate on which statistical tests are conducted. PE corresponds to

Prediction errors collapsed across reward and punishment conditions in Fig. 3 (GLM1)
Either reward or punishment PE in Fig. 4.

To quantify the number of recorded sites related to prediction errors (PE) for each brain parcel (Supplementary Table 2), we assessed significance of regression estimates in the time domain by applying a correction for multiple comparison (focusing on the in the [0.25 1 s] time window) using the false discovery rate algorithm[33] and by discarding significant effects lasting <100 ms. Because using alternative methods to define the individual statistical threshold (FDR, computed as above, uncorrected $p < 0.005$ or a common threshold such that the average regression estimates had to be >0.1) yielded to qualitatively similar results, we used the common threshold for clarity/display purpose to generate Supplementary Fig. 5.

To assess the contribution of the different frequency bands to prediction errors, we used the following GLM:

$$PE = \beta_\gamma \times Y(\gamma) + \beta_\beta \times Y(\beta) + \beta_{e\theta\alpha} \times Y(e\theta\alpha) + \beta_{l\theta\alpha} \times Y(l\theta\alpha) \tag{7}$$

With $\beta_\lambda$, $\beta_\beta$, $\beta_{e\theta\alpha}$, and $\beta_{l\theta\alpha}$ corresponding to the regression estimates of the power time series Y in the broadband gamma, beta, early theta-alpha and late theta-alpha bands. This GLM was compared to the eight possible alternative GLMs that combine BGA power to a single other frequency band (beta or early theta-alpha or late theta-alpha), two additional frequency bands (beta and early theta-alpha or beta and late theta-alpha or early and late theta-alpha) or all possible frequency bands (beta and early theta-alpha and late theta-alpha).

The model comparison was conducted using the VBA toolbox (Variational Bayesian Analysis toolbox; available at http://mbb-team.github.io). Log-model evidence obtained in each recording site was taken to a group-level, random-effect, Bayesian model selection (RFX-BMS) procedure[74]. RFX-BMS provides an exceedance probability (Xp) that measures how likely it is that a given model is more frequently implemented, relative to all the others considered in the model space, in the population from which samples are drawn.

For the separate investigation of prediction error components, two separate analyses were conducted for reward and punishment PE. For each analysis, power time-series Y was regressed against both outcome (R or P) and expectation (Qr or Qp):

$$Y = \alpha + \beta_1 \times R + \beta_2 \times Q \tag{8}$$

With $\beta_1$ and $\beta_2$ corresponding to the outcome (R or P) and expectation (Qr or Qp) regression estimates.

For all GLMs, significance of regressors was assessed using one-sample two-tailed $t$-test. T-values and $p$-values of those tests are reported in the result section. Once regions of interest were identified, significance was assessed through permutation tests within each ROI. The pairing between power and regressor

values across trials was shuffled randomly 60,000 times. The maximal cluster-level statistics (the sum of $t$-values across contiguous time points passing a significance threshold of 0.05) were extracted for each shuffle to compute a 'null' distribution of effect size across a time window of $-3$ to 1.5 s around outcome onset. For each significant cluster in the original (non-shuffled) data, we computed the proportion of clusters with higher statistics in the null distribution, which is reported as the 'cluster-level corrected' $p_c$-value.

**Reporting summary**. Further information on research design is available in the Nature Research Reporting Summary linked to this article.

## Data availability
Raw data cannot be shared due to ethics committee restrictions. Intermediate as well as final processed data that support the findings of this study are available from the corresponding author (J.B.) upon reasonable request.

## Code availability
The custom codes used to generate the figures and statistics are available from the lead contact (JB) upon request.

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

## Acknowledgements

This work benefited from the program from University Grenoble Alpes, within the program 'Investissements d'Avenir' (ANR-17-CE37-0018; ANR-18-CE28-0016; ANR-13-TECS-0013) and from the European Union Seventh Framework Program (FP7/2007–2013) under (grant 604102, Human Brain Project). M.G. received a PhD fellowship from Region Rhône-Alpes (grant ARC-15-010226801). The funders had no role in study design, data collection, and analysis, decision to publish or preparation of the manuscript. We thank all patients; the staff of the Grenoble Neurological Hospital epilepsy unit; and Patricia Boschetti, Virginie Cantale, Marie Pierre Noto, Dominique Hoffmann, Anne Sophie Job and Chrystelle Mosca for their support.

## Author contributions

M.P. and J.B. designed the experiment. M.G. collected the data. A.L.P., J.P.L., and O.D. provided preprocessing scripts and anatomical location of iEEG sites. M.G., P.B., and J.B. performed the data analysis. P.K., L.M., and S.R. did the intracerebral investigation and allowed the collection of iEEG data. M.G., M.P., and J.B. wrote the manuscript. All the authors discussed the results and commented on the manuscript.

## Competing interests

The authors declare no competing interests.
