## [Peer Review File · Nature Communications]

REVIEWER COMMENTS

Reviewer #1 (Remarks to the Author):

This well-written and exciting paper by Bastin and colleagues uses a unique dataset of invasive human EEG data to investigate the presence of reward and loss prediction errors (PEs). This is a timely paper that nicely extends the large corpus of non-invasive human and invasive non-human animal research. However, several aspects of the analyses are unclear and need to be addressed before I can recommend this paper for publication.

- The PE effects seem to appear extremely early (e.g., 90ms after outcome onset in insula). This is surprising and seems to be even before a proper object recognition could have taken place. How is this possible? To my knowledge, earliest signals in visual cortex appear after ~50ms. This would mean that in only 40ms the brain is able to compute the complex PE calculations, which are likely multi-synaptic and may even involve dopamine signalling.

- The (first?) EEG analysis is somewhat unorthodox. The standard approach for such model-based EEG analyses (in line with standard fMRI practices) is to use the cognitive variable (here: PEs) as a predictor for the EEG data (dependent variable; e.g. Philiastides 2010; Hauser 2015). The motivation for this is not quite clear. It looks like the authors wanted the different frequency bands (which I assume are correlated) to compete for variance. This would mean that the authors ask the question 'are there unique contributions of each freq. band?' rather than 'does this freq. band encode PEs?'. I am not sure this was their goal and why. The different analyses and their meaning should be motivated and explained more clearly.

- I could not completely figure out how baseline correction was conducted, but it looks like the authors took the time before outcome presentation to do the correction. This is obviously problematic because in this phase a key aspect of the PE may already be in the signal. The expectation component is evoked during/before this time. A baseline correction could thus induce such an expectation effect in the outcome phase even though there may be no trace of expectation in the actual signal.

- Because outcome probabilities do not change within one session, the PEs are probably closely related to time (large PEs early in the session). This could also influence the EEG results (fatigue, etc) especially in such a patient population. Do the results remain when controlling for trial number?

- Can the authors dissociate the PEs from salience PEs (i.e. absolute PEs)? Because PEs are usually sampled in a highly biased manner (e.g. few negative PEs in reward learning), these variables are usually closely related.

- In general, the different analyses must be detailed better in the results section. It is hard to understand whether and when PEs were analysed across both domains, when which analysis pipeline was used, etc.

- Do the authors have data on anterior cingulate/dorsomedial PFC? There is good animal data on PEs (e.g. Kennerley et al 2011). And human data would be most informative.

- How can the authors marry their findings with previous findings stating an absence of PEs in ventral striatum using similar techniques (Stenner et al 2014)? Do they have any recordings from VS?

Minor:

- Timing issues in human fMRI (line 80ff) are not a valid point. Plenty of papers have successfully separated value and PE/outcome phases using adequate jittering (e.g. Chowdhuri 2013 NN for example)

- The task should be described in the results section. For example, the outcome probabilities are only hidden in the methods section.

Reviewer #2 (Remarks to the Author):

In this paper, Gueguen et al. seek to disentangle the neural basis of reward prediction errors using intracranial recordings from epilepsy patients. The patients play a game in which they make decisions under uncertainty and have to learn to associate cues with rewards. The patients seem to be able to play the game, and authors model the patients' choice behavior using a QL model that accounts for the observed behavioral data well. The dataset in the paper is quite large for studies of this type ($n=20$), and the authors leverage this to their advantage examining neural activation and its association with reward-prediction errors (RPEs) in a variety of reward-related brain areas including among others OFC, LPFC and insula. The authors show that a simple regression model reveals a relationship between broadband high-gamma activity (BGA) and RPE. Power in lower frequency bands is also significantly associated with RPEs, but interestingly these associations do not have explanatory power beyond what BGA provides. Finally, the authors carry out additional analysis to disentangle the contribution of the expectation and outcome components of RPE, and report differential encoding of positive/negative RPE encoding across brain regions.

Overall, the results here reported are novel and interesting, and the task and analyses are well designed. The behavioral modeling is appropriate and carefully constructed, and the electrophysiological analyses are adequate for the questions at hand. The authors are careful to disentangle components of the RPE signal and the contribution of activity in different frequency bands to the RPE encoding. However, there are a few aspects of the paper that require improvement before being accepted. First, it is missing a fair amount of detail regarding the fact that these recordings are carried out in epilepsy patients (both from the behavioral and electrophysiological standpoints) that need to be included. In addition, some choices in the electrophysiological analysis require attention (see below). These are relatively minor concerns that the authors should be able to adequately address.

Main comments:

- The authors estimate BGA activity by averaging band-passed activity in 10Hz bands between 50 and 150Hz, as in previous studies from some of the same authors (Lopez-Persem Nat. Neurosci. 2020). They used a similar technique in lower frequencies (theta and alpha with 1Hz bins, and beta with 5Hz bins). I am not convinced that this is a valid analytical technique. Going on the assumption that the BGA reflects a single underlying neurobiological process (and the notion that BGA activity carries out a unique computational signal, supported by the results in this manuscript and others), a simpler method would be to use a single bandpass filter to estimate activity across the whole frequency band of interest (50-150Hz). Since there will be frequency bleed-through between adjacent 10Hz bins, and if furthermore those reflect the same underlying neural process, the argument could be made that this analysis amounts to averaging similar estimates of the same signal, a kind of bootstrapping or, at the very least, of frequency smoothing. This is defensible, and I invite the authors to do that in the text by explaining these nuances, but the best defense would be to present evidence as to whether this is the case. The authors should rerun their RPE analyses using a band-passed estimation for BGA and report whether the results are comparable to their original BGA estimation technique.
- It is unclear whether white matter electrodes, which do not pick up local neuronal activity like grey matter electrodes, were included or excluded from analysis – please clarify. If they were, how do their PE information encoding profiles compare to grey matter electrodes?
- One of the patients' recordings were taken at a 256Hz digitization rate. As per Nyquist, digitization rate must be at least twice the highest frequency of interest (and most people would recommend 3-4x). Thus, a digit rate of 256Hz is not enough to estimate frequencies above ~125Hz as included in the study. This patient should be excluded from all BGA analyses.
- What steps did the authors take to avoid contamination with electrophysiological artifacts? Specifically, how were electrodes close to the epileptic foci identified and removed from the analyses, and how were epochs containing epileptiform activity identified and removed from all other electrodes? This is an essential step for iEEG analysis – I suspect the authors may have removed it due to space constraints but it is important and must be included.
- As with all iEEG studies, the fact that patients have a brain pathology needs to be considered. In particular, how sure can we be that the patients' decision-making behavior is not impacted by their epilepsy diagnosis? The authors report (and Fig. 1b shows) that patients in the study achieve >70% choice accuracy, a reasonable threshold, but far from optional. How does this compare to healthy controls?
- As patients learn the task, they experience higher proportion of +RPEs than -RPEs. I wonder if these differences in trial numbers are impacting the regression estimates (Fig. 4), and may result in an underestimation of -RPE betas. This concern is alleviated in insula and dLLPFC, since -RPE signals are more prominent, but remains for the regions where +RPEs were higher (vmPFC and OFC). It would be nice to see the regression estimate comparison (barplots in Fig. 4c) in trial-matched conditions (i.e. by selecting a subset of +RPE trials to match the number of available -RPE trials).

Minor comments:

- This is by no means necessary, but if the authors were so inclined it would be nice to read further speculation on the origins of the asymmetry between + and -RPE signals, from the standpoint of neuromodulator systems (dopamine and serotonin) or otherwise.

- Fig.2: recommend matching color coding of electrodes (top) to regions (bottom) for legibility.
- Fig.2: recommend including electrode counts per region (top).
- Were the 3-6 blocks carried out in the same session, or across multiple ones?
- Fig.4a: I assume this is a typo – legend should specify +RPE or -RPE (right now it just reads RPE for both traces).

Reviewer #3 (Remarks to the Author):

Gueguen et al. investigated the neural correlates of prediction errors in the human brain using iEEG during an instrumental learning task. The task consists of pairs of cues, for which subjects had to learn to identify the stimulus that provided reward (+1) or that avoided a punishment (0 reward). The principle observation is that the broadband-gamma power measured with electrodes in the ROIs of interest (insula, dlPFC, vmPFC, IOFC) correlated positively with PE as well as outcome. In some areas, this correlation was stronger for PE during rewarding pairs, whereas it was stronger in the conditions in which subjects learned to avoid punishment. The overall conclusion drawn from this is that there is an ‘opponent’ coding of PE in the two conditions, with vmPFC+IOFC encoding PE more strongly during the reward PE condition, and insula and DLPFC encoding PE more strongly during the punishment reward condition. Also, in all four ROIs, gamma band power was correlated positively with outcome and negatively with expectation.

This study is interesting and takes advantage of the rare ability to record intracranially in humans. The analysis is sophisticated and includes state-of-the art RL-based modelling of behavior.

However, it isn’t clear to me what novel conceptual insight the conclusions made provide and how these are novel with respect to prior literature, including that with iEEG (i.e. Ramayya et al, Neuroimage; Saez et al 2018 Curr Bio;) and more broadly the fMRI and macaque single-neuron literature. It is known from a variety of techniques, in particular recordings in macaques, that neurons in these areas encode different aspects of prediction errors, expectation, and outcome. Sophisticated ways have been developed to tease apart these variables and their relationship, but this is not done here. The observed signals are not linked to choices or learning, leaving it unclear to what extent these are related to behavior in the task. While it is interesting and important to confirm that these variables are represented at the gamma-band level in humans, it is not clear to me what novel is learned from this data.

Major issues:

1. It isn’t clear to me how this work shows ‘opponent’ coding: all 4 ROIs investigated correlated positively with PE, but some more strongly for the reward and some for the punishment PE. In what sense is this ‘opponent’? It instead seems to be a matter of degree, some areas (on average) care more about one PE vs the other. The task seems ill suited to investigate PE, since the outcomes are binary (+1, 0, -1). A proper PE signal should show a graded and signed response with outcome magnitude.

2. Contribution of single electrodes. The approach of considering all electrodes in an ROI together leaves it unclear how these results look at a per-electrode level, essentially eliminating the high spatial resolution of iEEG (instead doing essentially “fMRI” with iEEG). Do some electrodes only encode outcome, others only expectation, others only PE? At the single neuron level it is clear that the variables investigated are intermixed, so taking an “on average” view of an entire brain area leaves a lot of interesting signals unstudied. The strategy used leaves it unclear whether the differences between the areas is attributable to a change in the balance between “penalty” and “reward” sensitive electrodes or whether truly there is a mixture of these signals at the single-electrode level.

3. Distinction between expectation, outcome, and prediction error signals. The analysis of expected value is done after onset of the outcome. But at that point, the signals related to expectation are intermixed with outcome and prediction error. The expected outcome would be expected to be encoded before onset of the outcome, but this period is not utilized for analysis. The task has a delay period before onset of the outcome, so it isn't clear to me why this period is not used to identify outcome signals.

4. The abstract claims that “...opponent systems ... mediate the repetition of rewarded choices and the avoidance of punishment choices”. But no link between the PE signals and choices is made here, so this is not warranted. Also I would advise authors not to intermix terms. ‘PE’ and ‘surprise’ are used interchangeably. The definition of ‘outcome’ is often unclear, i.e. is this a binary variable (0 vs |1|) or a three-level variable?

5. Methods. Aspects of what was done were unclear:

5.1 when averaging activity across an ROI, is this average across all electrodes in that area or only the significant electrodes? i.e. the n of figs 3+4 is unclear to me.

5.2 it was unclear to me what this statement in the Method means - “Note that punishment PE were inverted to allow an easier comparison with reward PE” . This also applies to Fig 3 ? In the paper, it is mentioned multiple times that the correlation of PE with gamma band power is positive, so this statement is confusing.

6. Why were the punishment and reward pairs randomly intermixed in the task? It seems that this way subjects also need to learn of what kind a pair is to separately engage a ‘punishment’ or a ‘reward’ PE system. In case of a ‘0’ outcome they really wouldn't know unless they learned what a pair is. But if they know what a pair is they wouldn't make an error. Given this the ‘0’ outcomes are likely mostly early on when learning hasn't occurred yet? Overall this design calls for analysis of the emergence of these differential signals as a function of time in the task. It is possible that the different encoding weights are due to early vs late trials rather than different learning systems.

Point-by-point responses to reviewers

Reviewer #1

Remarks to the authors

This well-written and exciting paper by Bastin and colleagues uses a unique dataset of invasive human EEG data to investigate the presence of reward and loss prediction errors (PEs). This is a timely paper that nicely extends the large corpus of non-invasive human and invasive non-human animal research. However, several aspects of the analyses are unclear and need to be addressed before I can recommend this paper for publication.

We thank the Reviewer for the supportive comments. Following the suggestions, we clarified the result section and performed new data analyses that strengthen the main conclusion of the paper. We now demonstrate that (i) expectation signals do not depend on the baseline correction procedure and that (ii) signed prediction error signals are observed in all regions of interest on top of other factors such as trial index or absolute prediction error.

Major comments

1. The PE effects seem to appear extremely early (e.g., 90ms after outcome onset in insula). This is surprising and seems to be even before a proper object recognition could have taken place. How is this possible? To my knowledge, the earliest signals in the visual cortex appear after ~50ms. This would mean that in only 40ms the brain is able to compute the complex PE calculations, which are likely multi-synaptic and may even involve dopamine signaling.

Latencies are difficult to estimate with precision, due to the cluster-based permutation tests applied to the data (Sassenhagen and Draschkow, 2019). Yet it is correct that in BGA, the PE signals appeared quite early in the alns (around 100 ms) and later in other regions like the IOFC (around 400 ms). This is consistent with the response latencies of neurons in several reward-related brain regions (Bayer and Glimcher, 2005; Kobayashi et al., 2006; O'Neill and Schultz, 2010; Schultz, 2016; Schultz et al., 1997). Thus, the PE signals detected through intracranial BGA share timing properties with single-cell responses. In a predictive coding framework, the information about expectation is already present in brain activity when the outcome is delivered. As there are only two non-zero outcomes to expect, they can be identified in much shorter time than new visual stimuli, and making the difference between outcome and expectation (i.e. excitatory and inhibitory inputs) is something that neurons can do in 10 ms.

2. The (first?) EEG analysis is somewhat unorthodox. The standard approach for such model-based EEG analyses (in line with standard fMRI practices) is to use the cognitive variable (here: PEs) as a predictor for the EEG data (dependent variable; e.g. Philiastides 2010; Hauser 2015). The motivation for this is not quite clear. It looks like the authors wanted the different frequency bands (which I assume are correlated) to compete for variance. This would mean that the authors ask the question ‘are there unique contributions of each freq. band?’ rather than ‘does this freq. band encode PEs?’. I am not sure this was their goal and why. The different analyses and their meaning should be motivated and explained more clearly.

We agree with the Reviewer about the standard approach: this is indeed the approach that we took in all main analyses, with the cognitive variable (PE) as the predictor and iEEG activity as the dependent variable. The reverse analysis, with activity in the different frequency bands as predictors and the cognitive variable (PE) as a dependent variable, was a follow-up testing whether there was additional information in low-frequency bands compared to BGA. We already employed this complementary approach, following the standard analysis, in our previous paper (Lopez-Persem et al., 2020). We believe both the standard and the follow-up analyses are useful, as they address different questions: (1) which brain regions in which frequency bands do represent PE and (2) whether the different frequency bands represent independent information about PE.

The starting point of all analyses was the assumption, suggested by several experts in the field, that BGA is a neural index that bridges fMRI and spiking activity (Mukamel et al., 2005; Niessing, 2005; Ray et al., 2008; Rich and Wallis, 2017). To check that the a priori focus on BGA was justified in our study, we extracted mean power in the post-outcome time window, for each trial and frequency band, separately. We found significant PE representation not only in BGA but also in lower frequency bands. In order to assess whether activity in low frequency bands was adding any information about PE, we conducted a Bayesian model comparison between GLMs containing different combinations of frequencies as regressors. Among all possible combinations, the model that best explained PE was the one only including BGA.

This is an important result that justifies the focus on BGA, since there was no additional information (about PE) in low-frequency bands. It shows that, even if the different frequency bands are mathematically separable, they do not necessarily carry independent information.

We updated the whole “*iEEG: PE signals across ROIs and frequency bands*” result section to make it easier to read and to explicitly motivate each analysis.

3. I could not completely figure out how baseline correction was conducted, but it looks like the authors took the time before the outcome presentation to do the correction. This is obviously problematic because in this phase a key aspect of the PE may already be in the signal. The expectation component is

evoked during/before this time. A baseline correction could thus induce such an expectation effect in the outcome phase even though there may be no trace of expectation in the actual signal.

We agree this is a critical point. We applied baseline correction to BGA time-series by subtracting the average BGA value (and dividing by the standard deviation) computed over a large epoch [-3 to 3 s] around outcome onset. This procedure was useful to normalize BGA across trials. Yet we agree that this procedure could have biased the expectation effect that we observed after the outcome.

We thus re-analyzed BGA by normalizing time series across trials using baseline activity averaged over the 500ms fixation epoch. This new baseline correction procedure did not impact the results: a negative encoding of expected value was still clearly visible after outcome onset in all ROIs (see Figure S6), and the main conclusions regarding the difference of sensitivity to RPE versus PPE between ROIs remained unchanged.

We added a sentence in the method section to explain how BGA was normalized and we also added a sentence indicating that using another normalization procedure (i.e., baselining the signal with respect to the fixation epoch) yielded to identical conclusions.

Methods ('Electrophysiological analyses'):

For each trial, baseline correction (Z-score) was applied using the average and standard deviation computed over a 6s epoch centered around outcome onset (-3 to 3 s).

Results ('iEEG: breaking PE into outcome and expected signals'):

Because baseline correction might affect the decrease in BGA observed with expected reward or punishment, we checked that the results were robust (Figure S6) to a change in the time window used to extract baseline activity (500ms pre-fixation epoch instead of 6s around outcome onset).

Figure S6. Decomposed PE signals into outcome and expected value by domains (RPE :R and Qr; PPE: P and Qp) after using an alternative method to normalize of BGA. Averaged regression estimates obtained from a linear from linear fit of BGA with PE components modeled separately for the reward (R or Qr: blue) or punishment (P or Qp: red) condition. Regression estimates were averaged over a 0.5-.75 s time window. Asterisks indicate significance ($p < 0.05$, one-sample, two-tailed Student's t-test). n indicates the number of recording sites in each ROI. Error-bars correspond to inter-sites SEM. Notice that here,

BGA single-trial time-series were normalized using a 500 ms duration baseline epoch extracted during visual fixation preceding each cue-pair onset.

4. Because outcome probabilities do not change within one session, the PEs are probably closely related to time (large PEs early in the session). This could also influence the EEG results (fatigue, etc) especially in such a patient population. Do the results remain when controlling for trial number?

This is again an excellent point. Passage of time could indeed confound the comparison between RPE and PPE signals, since subjects get more punishments at the beginning and more rewards at the end of a learning session. To control for this confound, we included trial index in the GLM meant to explain BGA, which already contained two separate regressors for RPE and PPE. Results were unchanged after controlling for trial index: there was a moderate negative correlation between BGA and trial index in most ROIs, but this did not alter the difference in sensitivity to RPE versus PPE between ROIs (Figure S1).

We added a sentence in the result section to indicate that results remained unchanged after controlling for trial number and added a supplementary figure (Figure S1).

Results ('iEEG: comparison between reward and punishment PE'):

Second, we examined whether the functional dissociation could arise from a differential involvement in the different leaning phases, as patients typically get more punishments at the beginning and more reward at the end of a learning session. Also, during the course of leaning, patients could figure out the mapping between the different pairs of cues and reward versus punishment domains and hence reframe their expectations. We checked that the difference between RPE and PPE signals observed across the four regions of interest was still significant even after controlling for trial index within learning sessions (Figure S1).

Figure S1. Dissociation of reward PE (R-Qr) and punishment PE (P-Qp) signals after controlling for trial number. Averaged regression estimates obtained from a linear fit of BGA with PE and trial index modeled separately for the reward (RPE and TRIALS: blue) or punishment (PPE and TRIALS: red) condition. Regression estimates were averaged over a 0.25-.1 s time window. Asterisks indicate significance (white: one-sample, two-tailed Student's t-test; black: paired-samples, two-tailed Student's t-test). N indicates the number of recording sites in each ROI. Error-bars correspond to inter-sites SEM.

5. Can the authors dissociate the PEs from salience PEs (i.e. absolute PEs)? Because PEs are usually sampled in a highly biased manner (e.g. few negative PEs in reward learning), these variables are usually closely related.

It is correct that absolute and signed PE are correlated in our design, so it is difficult to dissociate between the two. We have tried a GLM including both signed and absolute PE, modeled separately for the reward and punishment conditions. Unsurprisingly, regression estimates were similar for signed and absolute PE in all ROIs (Figure R1). Note however that all results hold with this new GLM (signed PE were still significantly encoded and significantly different between reward and punishment conditions in all ROIs). Given the correlation with signed PE, we prefer not to draw any strong conclusion about the encoding of absolute PE. We do not think it can be univocally interpreted as an impact of salience, as salience is related to rarity of events. Yet, although the frequency of rewards and punishments varied across learning sessions, PE regression estimates were stable (see response to Reviewer 3). We have now added a cautionary note in the discussion to acknowledge the correlation between signed and absolute PE.

Figure R1. Averaged regression estimates (over a 0.25-.1 s time window post outcome onset) obtained from a linear fit of BGA with signed vs. absolute PEs modeled separately for the reward or punishment condition. Asterisks indicate significance ($p < 0.05$, one-sample, two-tailed Student's t-test)

6. In general, the different analyses must be detailed better in the results section. It is hard to understand whether and when PEs were analyzed across both domains, when which analysis pipeline was used, etc.

We apologize if there was a lack of clarity regarding how analyses were done. We edited the Results section to make it easier to follow.

7. Do the authors have data on anterior cingulate/dorsomedial PFC? There is good animal data on PEs (e.g. Kennerley et al 2011). And human data would be most informative.

Unfortunately, the dorsomedial PFC was not sufficiently sampled in this study to pop-out in the whole-brain analysis. We checked again in the data and found that only three recording sites (belonging to two participants) responded

to PE in dorsomedial PFC. Because of this low statistical power, we decided that these results were not consistent enough across patients to be included in the paper.

8. How can the authors marry their findings with previous findings stating an absence of PEs in ventral striatum using similar techniques (Stenner et al 2014)? Do they have any recordings from VS?

The ventral striatum was not recorded in our patients. Yet, we would have expected to observe PE signals in the VS, as they are usually reported in fMRI studies. Besides limitations mentioned in the discussion of Stenner J Neurophysiol 2015, we might speculate that the absence of significant PE signals might relate to the large frequency band (30-150 Hz) including the beta range (and not just BGA), or the small number of patients (n=5) compared to our larger sample (n=20), or the use of a gambling task involving no PE-driven learning.

Minor comments

- Timing issues in human fMRI (line 80ff) are not a valid point. Plenty of papers have successfully separated value and PE/outcome phases using adequate jittering (e.g. Chowdhuri 2013 NN for example)

This was actually our point: without adequate jittering, it is difficult to identify PE because positive and expectation signals cancel each other. We agree that some fMRI papers successfully separated cue value and outcome PE, as we acknowledge in the introduction (with citation of Chowdhuri Nature Neuroscience 2013). Yet we still believe that most fMRI studies did not use adequate jittering, for understandable reasons (delays prolong the task and introduce memory issues).

Introduction

The issue can be solved by adequate jittering between cue and outcome events (Behrens et al., 2008; Chowdhury et al., 2013), but this has not been systematically used in human fMRI studies.

- The task should be described in the results section. For example, the outcome probabilities are only hidden in the methods section.

We apologize for the lack of methods details in the results section. We have now specified task description at the beginning of the Results section.

iEEG data were collected from twenty patients with drug-resistant epilepsy (see demographical details in Table S1 and methods) while they performed an instrumental learning task. Patients had to choose between two cues to either maximize monetary gains (for reward cues) or minimize monetary losses (for punishment cues). The pairs of cues associated to reward and punishment learning were intermingled within three to six sessions of 96 trials. In each pair, the two cues were associated to the two possible outcomes (0/1€ in the reward condition and 0/-1€ in the punishment condition) with reciprocal probabilities (0.75/0.25 and

0.25/0.75). Reward and punishment conditions were matched in difficulty, as the same probabilistic contingencies were to be learned.

Reviewer #2

Remarks to the authors

In this paper, Gueguen et al. seek to disentangle the neural basis of reward prediction errors using intracranial recordings from epilepsy patients. The patients play a game in which they make decisions under uncertainty and have to learn to associate cues with rewards. The patients seem to be able to play the game, and authors model the patients' choice behavior using a QL model that accounts for the observed behavioral data well. The dataset in the paper is quite large for studies of this type (n=20), and the authors leverage this to their advantage examining neural activation and its association with reward-prediction errors (RPEs) in a variety of reward-related brain areas including among others OFC, LPFC and insula. The authors show that a simple regression model reveals a relationship between broadband high-gamma activity (BGA) and RPE. Power in lower frequency bands is also significantly associated with RPEs, but interestingly these associations do not have explanatory power beyond what BGA provides. Finally, the authors carry out additional analysis to disentangle the contribution of the expectation and outcome components of RPE, and report differential encoding of positive/negative RPE encoding across brain regions.

Overall, the results here reported are novel and interesting, and the task and analyses are well designed. The behavioral modeling is appropriate and carefully constructed, and the electrophysiological analyses are adequate for the questions at hand. The authors are careful to disentangle components of the RPE signal and the contribution of activity in different frequency bands to the RPE encoding. However, there are a few aspects of the paper that require improvement before being accepted. First, it is missing a fair amount of detail regarding the fact that these recordings are carried out in epilepsy patients (both from the behavioral and electrophysiological standpoints) that need to be included. In addition, some choices in the electrophysiological analysis require attention (see below). These are relatively minor concerns that the authors should be able to adequately address.

We thank the Reviewer for the supportive comments and the helpful suggestions. We now provide more detailed explanations regarding the studied population of patients and how this might have impacted behavioral and electrophysiological analyses. We also clarified that the main result of the paper is more related to a difference of sensitivity across ROIs regarding the domain (reward vs. punishment) of prediction errors rather than their sign (positive or negative).

Main comments

1. The authors estimate BGA activity by averaging band-passed activity in

10Hz bands between 50 and 150Hz, as in previous studies from some of the same authors (Lopez-Persem Nat. Neurosci. 2020). They used a similar technique in lower frequencies (theta and alpha with 1Hz bins, and beta with 5Hz bins). I am not convinced that this is a valid analytical technique. Going on the assumption that the BGA reflects a single underlying neurobiological process (and the notion that BGA activity carries out a unique computational signal, supported by the results in this manuscript and others), a simpler method would be to use a single bandpass filter to estimate activity across the whole frequency band of interest (50-150Hz). Since there will be frequency bleed-through between adjacent 10Hz bins, and if furthermore those reflect the same underlying neural process, the argument could be made that this analysis amounts to averaging similar estimates of the same signal, a kind of bootstrapping or, at the very least, of frequency smoothing. This is defensible, and I invite the authors to do that in the text by explaining these nuances, but the best defense would be to present evidence as to whether this is the case. The authors should rerun their RPE analyses using a band-passed estimation for BGA and report whether the results are comparable to their original BGA estimation technique.

We understand that, at first sight, it may appear simpler to extract BGA through a single bandpass filter between 50 and 150 Hz and this is indeed done by other research groups in the literature. Yet, variations in BGA (i.e., a relative increase or decrease in power) around 100 or 140 Hz are likely to have the same functional importance as a change around 60 or 80 Hz. Therefore, these changes should be equally weighted in the broadband gamma index. Thus, the proposition of the reviewer to rerun all BGA analyses using a single bandpass filter would have substantially inflated the relative contribution of iEEG activity in the lower bound of BGA, while decreasing the contribution of higher frequencies (given the 1/f power law). Since our aim was to isolate relative changes in power within the broadband gamma around cognitive events, we think that our approach - filtering several sub-bands - provides a more robust estimation. This procedure was used multiple times in previous iEEG papers from our group and others (e.g., (Bastin et al., 2017, 2013; Lachaux et al., 2012, 2007; Ossandón et al., 2012; Perrone-Bertolotti et al., 2020).

Nevertheless, since we acknowledge that other groups use the single band-pass filtering approach, we rerun our RPE vs. PPE analyses using this method. It was reassuring to see that results were comparable to our original BGA estimation technique (Figure S4).

We expanded the paragraph that explained how BGA was estimated in the Methods section ('Electrophysiological analyses').

Finally, the envelope signals computed for each consecutive frequency bands (e.g., 10 bands of 10 Hz intervals between 50 and 150 Hz) were averaged together, to provide one single time-series (the BGA) across the entire session, expressed as percentage of the mean. This procedure was used to counteract a bias toward the lower frequencies of the frequency interval induced by the 1/f drop-off in amplitude. Finally, these time-series were smoothed with a 250 ms sliding window to increase statistical power for inter-trial and inter-individual analyses of BGA dynamics. This

procedure was previously shown to maximize the signal/noise ratio to detect task-related modulation of BGA by effectively smoothing the signal across frequencies and time to detect BGA modulations that exhibit across trials and across sites variability in terms of precise timing and frequency signature (Lopez-Persem et al., 2020; Bastin et al., 2017; Lachaux et al., 2007; 2012). In addition, we also checked whether estimating BGA using a single bandpass filter across the whole frequency band of interest (Saez et al., 2018) instead of filtering across consecutive frequency bands yielded to a similar pattern of result (Figure S4).

We also added a sentence in the Results section ('iEEG: comparison between reward and punishment PE') to specify that results were robust to methods used to compute or normalize BGA.

To confirm this dissociation between reward PE and punishment PE signaling regions, we performed a number of control analyses, at different levels from iEEG data preprocessing to model-based regressions. First, the functional dissociation was unchanged when using an alternative procedure for extracting BGA (See Methods and Figure S4) or after removing recording sites with pathological activity based on stringent exclusion criteria (See Methods and Figure S5).

Figure S4. Dissociation of reward PE (R-Qr) and punishment PE (P-Qp) signals using a single bandpass filter to estimate BGA. Time course of regression estimates obtained from linear fit of BGA with PE modeled separately for the reward (blue) and punishment (red) conditions (PPE: punishment prediction error; RPE: reward prediction error). Horizontal bold lines indicate significant difference between conditions (blue: RPE>PPE; red: PPE>RPE; $p < 0.05$). Shaded areas represent inter-sites SEM.

2. It is unclear whether white matter electrodes, which do not pick up local neuronal activity like grey matter electrodes, were included or excluded from analysis – please clarify. If they were, how do their PE information encoding profiles compare to grey matter electrodes?

We only included grey matter electrodes in the analyses, since the parcellation scheme we used for labeling contacts excluded white matter electrodes. Note that we could not compare grey vs. white matter electrodes because there were very few contacts in the white matter with completed recordings (i.e., these contacts were physically inside the brain but dismissed at the recording stage, only contacts localized in the grey matter were

selected for recording because our amplifier did not have the capacity to record activity from all available pairs of contacts in every patient).

We clarified this issue in the methods section ('Electrophysiological analyses').

Each recording site (i.e., each bipolar derivation) was thus labeled according to its location in a parcellation scheme in the patients' native space. Thus, the analyzed dataset only included electrodes identified to be in the grey-matter.

3. One of the patients' recordings were taken at a 256Hz digitization rate. As per Nyquist, digitization rate must be at least twice the highest frequency of interest (and most people would recommend 3-4x). Thus, a digit rate of 256Hz is not enough to estimate frequencies above ~125Hz as included in the study. This patient should be excluded from all BGA analyses.

We thank the Reviewer for pointing this out. Obviously we did not use the iEEG data from this patient to compute BGA. We only kept the behavioral data of this patient for statistical analyses.

This information is now reported in the methods section ('iEEG data acquisition and preprocessing'):

Neuronal recordings were conducted using an audio–video-EEG monitoring system (Micromed, Treviso, Italy), which allowed simultaneous recording of 128 to 256 depth-EEG channels sampled at 256 Hz (1 patient, note that this patient was removed from analyses based on broadband gamma activity which could not be computed given the low sampling rate), 512 Hz (6 patients) or 1024 Hz (12 patients) [0.1–200 Hz bandwidth].

4. What steps did the authors take to avoid contamination with electrophysiological artifacts? Specifically, how were electrodes close to the epileptic foci identified and removed from the analyses, and how were epochs containing epileptiform activity identified and removed from all other electrodes? This is an essential step for iEEG analysis – I suspect the authors may have removed it due to space constraints but it is important and must be included.

We agree this point should be addressed more carefully. However, since the results rely on model-based analyses of iEEG data, it is unlikely that they could be induced by noise or artifacts (the chance that epileptiform activity correlates across trials correlate with the computational variables of interest is virtually null). Thus, including all the iEEG data in the analysis, without selecting good recordings, could only play against the reported results. In other words, observing significant results despite using raw and noisy data is actually a guarantee of robustness. Furthermore, noise or artefacts cannot explain the functional dissociation between aINS-dIPFC being more sensitive to PPE than and vmPFC-IOFC more sensitive to RPE.

That being said, we acknowledge that it is common in the field to exclude pathological data and artefacts. We thus re-analyzed the whole dataset after applying the following exclusion criteria to the data: we removed (1) noisy (bad) channels, (2) recording sites showing a large proportion of epileptic spikes and (3) sites that were identified to be within the epileptic zone in operated patients. As expected (figure S5), the pattern was virtually unchanged (the same comparisons were statistically significant), but the correlation with computational variables were much better (all regression estimates were about doubled).

We added a paragraph in the methods ('iEEG data acquisition and preprocessing') and in the results ('iEEG: comparison between reward and punishment PE') to explain how artefacts and pathological activity were controlled in the study and why we chose to keep all channels in the main analyses.

Methods

In order to take advantage of our large sample set, all recording sites with an anatomical label were included in the analyses (i.e., without any exclusion of sites with artifacts or pathological activity). Nevertheless, to check whether electrodes with artifacts or pathological activity could have biased the results, we applied a semi-automatic pipeline: first, bad channels detection was conducted with a machine learning approach (Tuyisenge et al., 2018), secondly, epileptic spikes were detected automatically with Delphos - Detector of ElectroPhysiological Oscillations and Spikes – (Roehri et al., 2016) and all data were then finally visually inspected to check their quality. We also excluded recording sites that were part of the epileptogenic zone by identifying with the neurologists (PK and SR) all recording sites involved at seizure onset and/or sites that were located within the cortical resection (if any) performed after the sEEG. The results were not affected by this procedure..

Results

To confirm this dissociation between reward PE and punishment PE signaling regions, we performed a number of control analyses, at different levels from iEEG data preprocessing to model-based regressions. First, the functional dissociation was unchanged when using an alternative procedure for extracting BGA (See Methods and Figure S4) or after removing recording sites with pathological activity based on stringent exclusion criteria (See Methods and Figure S5).

Figure S5. Dissociation of reward PE (R-Qr) and punishment PE (P-Qp) after excluding recording sites exhibiting pathological activity. Time course of regression estimates

obtained from linear fit of BGA with PE modeled separately for the reward (blue) and punishment (red) conditions (PPE: punishment prediction error; RPE: reward prediction error). Horizontal bold lines indicate significant difference between conditions (blue: RPE>PPE; red: PPE>RPE; $p < 0.05$). Shaded areas represent inter-sites SEM. Notice the lower numbers of recording sites for all ROIs, following the stringent exclusion criteria applied to the data (see methods).

5. As with all iEEG studies, the fact that patients have a brain pathology needs to be considered. In particular, how sure can we be that the patients' decision-making behavior is not impacted by their epilepsy diagnosis? The authors report (and Fig. 1b shows) that patients in the study achieve >70% choice accuracy, a reasonable threshold, but far from optional. How does this compare to healthy controls?

We do not have data in matched healthy controls, but the success rate in our patients (71%) is actually very close to that reported (74%) in a previous study (Palminteri Nat Communications 2015) using a similar learning task (with the same cue-outcome contingencies). Thus, despite an older age, drug-resistant epilepsy and different testing conditions, our patients' behavior was in the normal range. This was already mentioned in the results section, we have now added the actual success rates for comparison with healthy controls. We also address this point in the revised discussion:

The data have been collected in patients being treated for epilepsy. It is unlikely that disease or treatment affected learning performance because the behavior of our patients was comparable to that of healthy young participants performing a similar task (e.g., Palminteri et al. 2015). We made the assumption that epileptic activity did not distort the brain signals linked to PEs and interpreted the data as if they were collected in healthy individuals. Furthermore, epileptic artifacts are unlikely to covary with the computational variable (PE) against which brain activity was regressed. We formally verified that manually removing artifacts did not affect the results about PE signals observed in BGA. We suggest that artifact removal may be unnecessary in model-based analyses of iEEG activity, and that results may actually be more robust if based on the entire dataset without ad-hoc selection based on visual inspection.

6. As patients learn the task, they experience higher proportion of +RPEs than -RPEs. I wonder if these differences in trial numbers are impacting the regression estimates (Fig. 4), and may result in an underestimation of -RPE betas. This concern is alleviated in insula and dlPFC, since -RPE signals are more prominent, but remains for the regions where +RPEs were higher (vmPFC and OFC). It would be nice to see the regression estimate comparison (barplots in Fig. 4c) in trial-matched conditions (i.e. by selecting a subset of +RPE trials to match the number of available -RPE trials).

We think there is a misunderstanding here, probably due to a lack of clarity on our part. We have not compared between +RPE and -RPE (PE of opposite sign) but between RPE and PPE (PE of different domain). As reward and punishment cues were interleaved, trials were actually matched between the two conditions, such that regression estimates in Fig. 4 are not confounded with trial index (see also Figure S1). We have now clarified throughout the

manuscript that opponent regions were differently sensitive to PE in different domains (reward versus punishment) and not different signs.

However, because Reviewer 3 also raised a concern about how PE signals varied across trials, we checked that PE signals remained stable between subsets of learning sessions (reviewer 3, comment 6, Figure S2).

Figure S2. Stability of contrasts between punishment (reward) and non-punishment (non-reward) outcomes in aINS and dIPFC (vmPFC and IOFC) over the course of learning. Average broadband gamma responses to relative punishments (i.e., the average of the difference of BGA between -1 € and 0€ outcomes) or to relative rewards (+1 € vs. 0€) split as a function of learning phase (early vs late trials). Early trials corresponded to the first 8 trials; Late trials corresponded to the last 8 trials. Asterisks indicate significance (one-sample, two-tailed Student's t-test).

Minor comments

7. This is by no means necessary, but if the authors were so inclined it would be nice to read further speculation on the origins of the asymmetry between + and -RPE signals, from the standpoint of neuromodulator systems (dopamine and serotonin) or otherwise.

This is an interesting point. Again the dissociation is not about + and -RPE signals but between RPE and PPE signals. The anatomical separation between our ROI was already discussed in detail.

We now discuss the possibility that the anatomical divide might relate to the projections of different neuromodulator systems (dopamine for RPE and serotonin for PPE).

The dissociation observed here at the cortical level between reward versus punishment PE signals might be related to afferences from different neuromodulatory systems, such as dopaminergic versus serotonergic pathways, which have been suggested to play opponent roles in approach versus avoidance behaviors (Daw et al. 2002; Boureau and Dayan; 2011).

8. Fig.2: recommend matching color coding of electrodes (top) to regions (bottom) for legibility.

9. Fig.2: recommend including electrode counts per region (top).

We thank the Reviewer for these helpful suggestions. We modified the color

code of the parcellation in Figure 2b to match the color code for electrode location in Figure 2a. We also augmented the legend with the number of recording sites per region.

Figure 2. Anatomical locations of intracerebral sites. **a.** Sagittal and axial slices of a brain template over which each dot represents one iEEG recording site (n=1694). Color code indicates location within the four regions of interest (red: vmPFC, n=54; green: dlPFC, n=74; blue: IOFC, n=70; purple: aINS, n=83). **b.** Parcellation scheme (adapted from MarsAtlas) represented on an inflated cortical surface.

10. Were the 3-6 blocks carried out in the same session, or across multiple ones?

All blocks were carried out in the same session, with a total duration spanning from 45min to 1h30 (including training and breaks of about 5 minutes between blocks).

We updated the methods section ('Behavioral task') to specify this point:

During iEEG recordings, patients performed three to six test sessions on a single testing occurrence (with short breaks between sessions).

11. Fig.4a: I assume this is a typo – legend should specify +RPE or -RPE (right now it just reads RPE for both traces).

This was not a typo. In Figure 4, the legend distinguishes between reward prediction error (RPE: $R-Q_r$) and punishment prediction error (PPE: $P-Q_p$).

To avoid any further confusion, we clarified the figure legend and modified the description of the results presented in Figure 4.

In the following analyses, we focused on BGA and tested whether prediction errors estimated in the reward (RPE= $R-Q_r$) and punishment (PPE= $P-Q_p$) conditions could be dissociated between the four ROIs previously identified (aINS, dlPFC, vmPFC and IOFC).

Reviewer #3

Remarks to the authors

Gueguen et al. investigated the neural correlates of prediction errors in the human brain using iEEG during an instrumental learning task. The task consists of pairs of cues, for which subjects had to learn to identify the stimulus that provided reward (+1) or that avoided a punishment (0 reward). The principle observation is that the broadband-gamma power measured with electrodes in the ROIs of interest (insula, dlPFC, vmPFC, IOFC) correlated positively with PE as well as outcome. In some areas, this correlation was stronger for PE during rewarding pairs, whereas it was stronger in the conditions in which subjects learned to avoid punishment. The overall conclusion drawn from this is that there is an 'opponent' coding of PE in the two conditions, with vmPFC+IOFC encoding PE more strongly during the reward PE condition, and insula and dlPFC encoding PE more strongly during the punishment reward condition. Also, in all four ROIs, gamma band power was correlated positively with outcome and negatively with expectation. This study is interesting and takes advantage of the rare ability to record intracranially in humans. The analysis is sophisticated and includes state-of-the-art RL-based modelling of behavior.

However, it isn't clear to me what novel conceptual insight the conclusions made provide and how these are novel with respect to prior literature, including that with iEEG (i.e. Ramayya et al, Neuroimage; Saez et al 2018 Curr Bio) and more broadly the fMRI and macaque single-neuron literature. It is known from a variety of techniques, in particular recordings in macaques, that neurons in these areas encode different aspects of prediction errors, expectation, and outcome. Sophisticated ways have been developed to tease apart these variables and their relationship, but this is not done here. The observed signals are not linked to choices or learning, leaving it unclear to what extent these are related to behavior in the task. While it is interesting and important to confirm that these variables are represented at the gamma-band level in humans, it is not clear to me what novel is learned from this data.

Let us first clarify that the observed brain signals in our study were associated across trials to prediction errors that were inferred from choice behavior through computational modeling. Thus, by construction, they were linked to how choices were shaped by successive outcomes. In other words, they represented the teaching signals that, under a common assumption (that the brain does implement some form of reinforcement learning), was driving choice behavior.

Yet we agree that tracking PE signals in the brain, even related to choice and learning, is not fundamentally novel. The novelty of our study is that we were able to disentangle the sign of prediction error (better or worse than expected) and the learning domain (reward-based approach versus punishment-based avoidance). Most studies confound reward or gain with better-than-expected outcome and punishment or loss with less-than-expected outcome. This is

typically the case of the papers cited by the reviewer (Ramayya et al. NeuroImage 2015 or Saez et al. Current Biology 2018), which used tasks with binary outcomes (positive or negative). As we intended to explain in the introduction, this is problematic because winning is confounded with not losing, and losing with not winning. We avoid this pitfall as in each learning domain (reward or punishment) we have both positive and negative outcomes (reward and no reward or punishment and no punishment). This is why we can conclude that the main divide in the brain is between regions signaling reward versus punishment PE and not positive versus negative PE. Thus, the fundamental divide is about opponent behaviors (approach versus avoidance learning) and not opposite valence (good versus bad outcomes).

Beyond this fundamental aspect, our study has other strengths that are quite rare to our knowledge, notably the use of oblique electrode trajectory that enables recording activity in the anterior insular lobe, which previous fMRI and lesion studies have identified as a key contributor to punishment-avoidance learning (e.g., Pessiglione et al., Nature 2006; Palminteri et al., Neuron 2012).

Thus, we do think that our findings bring novel and critical evidence regarding the dissociation between brain systems supporting reward versus punishment learning.

We also improved the paper following some of the Reviewer suggestions for additional analyses which are detailed in the point-by-point response below.

Major comments

1. It isn't clear to me how this work shows 'opponent' coding: all 4 ROIs investigated correlated positively with PE, but some more strongly for the reward and some for the punishment PE. In what sense is this 'opponent'? It instead seems to be a matter of degree, some areas (on average) care more about one PE vs the other. The task seems ill suited to investigate PE, since the outcomes are binary (+1, 0, -1). A proper PE signal should show a graded and signed response with outcome magnitude.

This is indeed an important point. We used 'opponent' as a shortcut but we acknowledge it may not be the best way to describe the difference between our ROIs. It follows on a tradition (Boureau and Dayan, 2011; Daw et al., 2002; Seymour et al., 2005) viewing brain regions tracking RPE versus PPE as opponent systems because they have opposite impact on the behavior (approach versus avoidance). We never meant that opponent systems show 'opponent coding', only that they are differently sensitive to RPE and PPE, as the Reviewer correctly noted. To avoid any confusion, we have removed the term 'opponent' from the title and abstract. We only use the term when specifically referring to the literature on opponent reward and punishment systems. We also checked that there was no mention of 'opponent coding' anywhere in the manuscript.

Regarding the task, we agree that varying the magnitude of outcomes could be a way to vary PE across trials. However, the key concept of PE is that the

response to outcome depends on expectation. This is why many studies, including the seminal papers in monkeys (e.g., Schultz et al., Science 1997), varied expectations but not outcome magnitude. It is also the case of studies in humans testing for dissociations between reward and punishment learning (e.g., Frank et al., Science 2004), on which our own learning task is built. Indeed, the same task was successful in showing the differential impact of dopaminergic drugs on RPE versus PPE (Pessiglione et al Nature 2006), or the differential impact of brain lesions on reward versus punishment learning (Palminteri et al Neuron 2012). Thus, while we agree that the task is not exhaustive in testing the different dimensions along which PE could vary, we disagree with the idea that it is not appropriate to dissociate RPE from PPE signals. In addition, it allows direct comparisons with fMRI, pharmacological and lesion studies using the same task to show the functional implication of the same ROIs.

We added a paragraph to describe the task at the beginning of the results section.

Patients had to choose between two cues to either maximize monetary gains (for reward cues) or minimize monetary losses (for punishment cues). The pairs of cues associated to reward and punishment learning were intermingled within three to six sessions of 96 trials. In each pair, the two cues were associated to the two possible outcomes (0/1€ in the reward condition and 0/-1€ in the punishment condition) with reciprocal probabilities (0.75/0.25 and 0.25/0.75). Reward and punishment conditions were matched in difficulty, as the same probabilistic contingencies were to be learned.

And (in the Results section 'Computational modeling'):

In the following analyses, iEEG activity was regressed against PE estimated for each participant at each trial in each condition from the QLr model fit. PE therefore represents our key independent variable – a hidden variable that in principle could have driven learning, since it is informed by individual choice behavior.

We also added a sentence in the discussion to better explain the advantage of this task relative to previous studies that compared neural signals related to the domain (reward vs. punishment) of prediction errors.

This anatomical dissociation between learning systems in the brain may appear at variance with previous studies reporting that rewards and punishments are ubiquitously represented all over the brain (Vickery et al., 2011; Ramayya et al., 2015). However, these non-specific results were observed during tasks in which the outcome is either reward or punishment. Thus, it is understandable that both reward and punishment regions were mobilized by the outcome in these previous studies, since being rewarded is not being punished and vice-versa.

2. Contribution of single electrodes. The approach of considering all electrodes in an ROI together leaves it unclear how these results look at a per-electrode level, essentially eliminating the high spatial resolution of iEEG (instead doing essentially "fMRI" with iEEG). Do some electrodes only encode outcome, others only expectation, others only PE? At the single neuron level it is clear that the variables investigated are intermixed, so taking an "on

average” view of an entire brain area leaves a lot of interesting signals unstudied. The strategy used leaves it unclear whether the differences between the areas is attributable to a change in the balance between “penalty” and “reward” sensitive electrodes or whether truly there is a mixture of these signals at the single-electrode level.

We do not share the view that an iEEG electrode is a meaningful spatial scale for assigning a cognitive function (note that spatial resolution is actually close to fMRI and far from single-cell recording). Yet we understand and respect the Reviewer’s request. Following on the suggestion, we now provide estimates of RPE and PPE signals for each recording site and Venn’s diagrams for all four main ROIs (Figure S3).

We have included the figure in the revised manuscript for the sake of transparency, but concluding on the balance between ‘penalty’ and ‘reward’ electrodes makes less sense to us. Labeling an electrode as ‘significant’ obviously depends on the (arbitrary) statistical threshold and neglects what should be important (the effect size), i.e. its contribution to the global signal. Having an electrode below threshold for reward and above threshold for punishment is no guarantee that there is true difference between conditions, which is why we prefer to test the difference (RPE-PPE). Besides, as a majority of electrodes do not pass significance threshold, reasoning on significant electrodes would be reasoning on tails of distributions instead of central tendencies. For these reasons we believe that the appropriate analysis, for assigning cognitive properties to a given ROI, should be based on the entire distribution of effect sizes across all electrodes.

Here, to set the threshold, we chose to use a common threshold across all electrodes to generate Figure S3 (i.e., average regression estimates above .1 were considered significant). Importantly, we checked that results were robust across the use of alternative methods to set the threshold: we tried to apply more conservative method by applying a FDR correction in the time domain for each electrode or less conservative method such as setting for each electrode the significance level to $\alpha=0.005$ with the additional constraint that significant effects had to last at least 100 ms. Critically, even if the absolute number of significant sites changed, the distribution across domains remained unchanged (i.e., there were always more sites responding to RPE than PPE in IOFC and vmPFC whereas the reverse pattern was observed in aINS and dlPFC).

We added a paragraph in the result section to report these new findings (and a supplementary Figure S3).

Results (‘iEEG: comparison between reward and punishment PE’):

Third, we examined whether the differences between ROIs signaling reward and punishment PE could arise from differences in the proportion of recording sites sensitive to reward versus punishment (Figure S3). In the aIns and dlPFC, recording sites were more likely to represent punishment prediction errors whereas in the vmPFC and IOFC, recording sites were more likely to represent reward prediction

errors.

Figure S3. Dissociation of reward and punishment PE signals across recording sites.

a. Average regression estimates obtained from linear fit of BGA with PE modeled separately for the reward and punishment conditions (PPE: punishment prediction error; RPE: reward prediction error) for all recording sites within each region of interest (estimates were averaged within the cluster in which the RPE vs. PPE contrast was significantly). Horizontal lines indicate average across sites (blue: RPE; red: PPE; black: RPE-PPE). n indicates the number of recording sites in each ROI. **b.** Venn diagram indicating the number of recording sites encoding RPE and/or PPE (or nothing).

3. Distinction between expectation, outcome, and prediction error signals. The analysis of expected value is done after onset of the outcome. But at that point, the signals related to expectation are intermixed with outcome and prediction error. The expected outcome would be expected to be encoded before onset of the outcome, but this period is not utilized for analysis. The

task has a delay period before onset of the outcome, so it isn't clear to me why this period is not used to identify outcome signals.

Stating that, at outcome onset, expectation signals are "intermixed" with outcome and prediction error signals makes things more confusing than they actually are: expectation signals are simply subtracted to outcomes to generate PE signal. Because of this linear combination, it is appropriate to use a general linear model (GLM) approach that estimates the influence of both PE components (outcome and expectation) on BGA following outcome onset.

Regarding expectation signals, we think that the Reviewer made a good point in pointing that we did not analyze the time window between the choice made by the participant and the display of outcome delivery. We have now extended our regression against expectation to this delay period (Figure S7) and included a paragraph in the results section to depict these new findings. In all ROIs, the correlation between BGA and expectation tended to be positive before the outcome and negative the outcome. Even in not always significant, this tendency corroborates the idea, that expectation is present in the signal before the outcome, and subtracted only when the outcome is delivered.

Results ('iEEG: comparison between reward and punishment PE'):

We next examined whether expectation signals could be observed during the delay period preceding outcome delivery (i.e., by averaging regression estimates over a -1.2 to -0.2 time window before outcome onset). We found that expected punishments (Qp) was positively represented in both aINS and dlPFC whereas although their mean were also positive in the vmPFC and IOFC, regression estimates were not significantly different from zero in the reward-sensitive ROIs (Figure S7).

Figure S7. Expectation signals during the delay vs. outcome phase. Average regression estimates obtained from linear fit of BGA with expectation modeled separately for the punishment condition (Qp: punishment expectation) or the reward condition (Qr: reward expectation). Time windows used to average regression estimates were time-locked to outcome onset: Delay time window corresponds to [-1.2 to -0.2 s]; Outcome time window corresponds to [.25 to 1s]. Stars indicate significance (one-sample, two-tailed Student's t-test). n indicates the number of recording sites in each ROI. Error-bars correspond to inter-sites SEM and dots correspond to individual recording sites.

We also briefly discussed this new finding in the discussion section:

The (positive) correlation with the outcome and the (negative) correlation with the expectation were simultaneously observed after outcome display. Interestingly, expectations tended to be positively associated with BGA, the sign of the correlation being reverted when the outcome was delivered. These observations, made possible here by the high temporal resolution of iEEG, are rarely reported in fMRI studies (Fouragnan et al., 2018). One reason is that the hemodynamic response, because of its low temporal resolution, may confound positive expectation at cue onset and negative expectation at outcome onset, unless the two events are separated by a long delay (as in, e.g., Behrens et al., 2008 or Chaudhuri et al., 2013).

4. The abstract claims that "...opponent systems ... mediate the repetition of rewarded choices and the avoidance of punishment choices". But no link between the PE signals and choices is made here, so this is not warranted. Also I would advise authors not to intermix terms. 'PE' and 'surprise' are used interchangeably. The definition of 'outcome' is often unclear, i.e. is this a binary variable (0 vs |1|) or a three-level variable?

We checked the manuscript and did not find why 'PE' and 'surprise' are said to be used interchangeably. On the contrary, these terms have precise meanings that we intended to respect: the element of surprise in PE is the absolute difference between outcome and expectation, and PE is just surprise with a sign, depending on whether the outcome is better or worse than expected.

Regarding the way outcome is modeled, since RPE and PPE are included in separate regressors, they can only be binary (0 or 1 for RPE, 0 or -1 for PPE).

We have made this more explicit (beginning of the Results section) to avoid the reader being confused:

iEEG data were collected from twenty patients with drug-resistant epilepsy (see demographical details in Table S1 and methods) while they performed an instrumental learning task. Patients had to choose between two cues to either maximize monetary gains (for reward cues) or minimize monetary losses (for punishment cues). The pairs of cues associated to reward and punishment learning were intermingled within three to six sessions of 96 trials. In each pair, the two cues were associated to the two possible outcomes (0/1€ in the reward condition and 0/-1€ in the punishment condition) with reciprocal probabilities (0.75/0.25 and 0.25/0.75). Reward and punishment conditions were matched in difficulty, as the same probabilistic contingencies were to be learned.

Finally, it is not true that there is no link between PE signals and choices made. The reason is that the RL model is fitted on choice behavior. So the PE time-series that were regressed against BGA are precisely those that, in principle, could have guided learning of reward and punishment value, have approach and avoidance choices. Now we agree that we have no proof that the brain is actually using the PE signal in the way the model does (we even acknowledge that neural processes responsible for choices are likely to be more complex), but this is a critique that could be addressed to any model-based analysis of neural data. We nonetheless added a cautionary note stating explicitly that one can only speculate about how the brain uses PE-

related information contained in BGA.

We also included a sentence in the result section to make it easier to understand and rephrased the incriminated sentence in the abstract.

Results ('Computational modeling')

In the following analyses, iEEG activity was regressed against PE estimated for each participant at each trial in each condition from the QLR model fit. PE therefore represents our key independent variable – a hidden variable that in principle could have driven learning, since it is informed by individual choice behavior.

Abstract (the term opponent was removed)

These regions might therefore belong to brain systems that differentially contribute to the repetition of rewarded choices and the avoidance of punished choices.

5. Methods. Aspects of what was done were unclear:

5.1 when averaging activity across an ROI, is this average across all electrodes in that area or only the significant electrodes? i.e. the n of figs 3+4 is unclear to me.

We clarified that in all analyses, all recording sites located within a given ROI were included: we did not select the sites depending on their response to task events.

5.2 it was unclear to me what this statement in the Method means - "Note that punishment PE were inverted to allow an easier comparison with reward PE". This also applies to Fig. 3 ? In the paper, it is mentioned multiple times that the correlation of PE with gamma band power is positive, so this statement is confusing.

We removed this statement and the manuscript was edited to clarify that positive reward prediction error corresponds to better than expected monetary gain whereas positive punishment prediction error corresponds to worse than expected monetary loss.

6. Why were the punishment and reward pairs randomly intermixed in the task? It seems that this way subjects also need to learn of what kind a pair is to separately engage a 'punishment' or a 'reward' PE system. In case of a '0' outcome they really wouldn't know unless they learned what a pair is. But if they know what a pair is they wouldn't make an error. Given this the '0' outcomes are likely mostly early on when learning hasn't occurred yet? Overall this design calls for analysis of the emergence of these differential signals as a function of time in the task. It is possible that the different encoding weights are due to early vs late trials rather than different learning systems.

As explained in the introduction, intermixing reward and punishment is crucial to obtain dissociations between approach and avoidance systems. This is a key insight that enabled clarifying the link between learning and dopamine in

the work of Michael Frank and colleagues (Frank Science 2004 and following papers). The reason is that, if distributed in separate blocks, outcomes are reframed such that not winning becomes punishing and not losing becomes rewarding. In other words, intermixing outcomes maintains the reference point at zero. It is not correct to state that participants 'need to learn of what kind a pair is to separately engage a punishment or reward system'. The task can be solved by tracking the average outcome of each cue separately, so as to select the one that is frequently rewarded and ignore the one that is frequently punished. Some participants never realize that the two cues of a pair always come together. It is reward and punishment outcomes that differentially engage opponent brain systems, which might then be recalled through associative learning at cue onset. We have extended our description of the task at the beginning of the results section to better clarify why reward and punishment pairs must be intermixed.

Nevertheless, we acknowledge that reframing could occur in the course of learning sessions if, as the Reviewer suggested, patients do realize which valence each pair is associated to. This would mean that PE are not calculated relative to each cue as in our RL model and that the engagement of reward and punishment systems would vary across trials. To test for this possibility, we performed a model-free analysis, simply contrasting zero and non-zero outcomes, at different stages of learning sessions (early and late trials). Results (Figure S2) show that contrast between reward and non-reward outcomes (following the reward cues) was stable in reward ROIs (vmPFC and IOFC) and that the contrast between punishment and non-punishment outcomes (following the punishment cues) was stable in punishment ROIs (alnS and dIPFC). Note that this is compatible with signaling PE as computed by our RL model, it simply neglects the expectation component (i.e. the component that varies with learning), because it cancels out when subtracting responses to outcomes. Indeed, if expectation is zero (no learning), the difference between RPE is theoretically 1 (+1€ for reward minus 0€ for no reward), while if the expectation is 0.75 (perfect learning), that difference is again 1 (+0.25€ for reward minus -0.75€ for no reward). The same reasoning holds for PPE, for which the difference between outcomes should also remain 1 throughout learning, whether the expectation is 0 or 0.75. The signals observed in our ROIs followed this pattern predicted by a model that tracks each cue-outcome association separately, without reframing in the course of learning.

We added a paragraph in the Results section ('iEEG: comparison between reward and punishment PE') to report these new findings (and a supplementary Figure S2).

Second, we examined whether the functional dissociation could arise from a differential involvement in the different leaning phases, as patients typically get more punishments at the beginning and more reward at the end of a learning session. Also, during the course of leaning, patients could figure out the mapping between the different pairs of cues and reward versus punishment domains and hence reframe their expectations. We checked that the difference between RPE and PPE signals observed across the four regions of interest was still significant even after controlling for trial index within learning sessions (Figure S1). We also checked that the

magnitude of PE signals was constant throughout the learning session. For this we simply used the contrast between possible outcomes (reward versus no reward and punishment versus no punishment), in a model-free analysis. In the RL framework, this contrast should be stable because it does not depend on what is learned (i.e., on expectations). Indeed, this contrast of BGA activity following zero and non-zero outcomes was of similar magnitude in early and late trials of learning sessions (Figure S2). This was observed in both reward PE (vmPFC and IOFC) and punishment PE (aINS and dlPFC) signaling ROIs. Thus, PE were reliably signaled from the beginning to the end of learning, with a stable difference between ROIs more sensitive to reward versus punishment outcomes.

Figure S2. Stability of contrasts between punishment (reward) and non-punishment (non-reward) outcomes in aINS and dlPFC (vmPFC and IOFC) over the course of learning. Average broadband gamma responses (over the 0.25-1 s time window) to relative punishments (i.e., the average of the difference of BGA between -1 € and 0€ outcomes) or to relative rewards (+1 € vs. 0€) split as a function of learning phase (early vs late trials). Early trials corresponded to the first 8 trials; Late trials corresponded to the last 8 trials. Stars indicate significance (one-sample, two-tailed Student's t-test). n indicates the number of recording sites in each ROI. Error-bars correspond to inter-sites SEM and dots correspond to individual recording sites.

REVIEWER COMMENTS

Reviewer #1 (Remarks to the Author):

The authors have thoroughly addressed all my concerns and I am happy to recommend this manuscript for publication.

Reviewer #2 (Remarks to the Author):

In their revision, the authors have mostly responded adequately to my concerns, with one important outstanding issue (see point #4 below). Let me touch on their responses:

1. BGA estimation technique (single bandpass vs. 10Hz bands). The authors make a reasonable point about the $1/f$ profile overweighting lower gamma power vs high-gamma power. Arguably, this could be compensated through baselining, but the point is valid. In any case, the authors' effort to carry the analyses with a single band method (the new figure S4) puts this point to rest altogether since the results are virtually identical. This is a satisfactory response.

2. Analyzing results from grey matter electrodes only is appropriate and the clarification is welcome.

3. This clarification is welcome as the distinction was not obvious in the original version of the paper.

4. The removal of noisy electrodes and associated explanation is welcome. However, the lack of exclusion of noisy epochs is troublesome. The authors make a reasonable claim that any artifactual activity is unlikely to explain the breadth of results. However, the claim that "including all the iEEG data in the analysis, without selecting good recordings, could only play against the reported results" is false. One can imagine a situation in which a single or few trials contaminated by large artifacts could be biasing (perhaps strongly) the regression estimates. The frequency with which this happens will depend on how noisy the recordings are (and there is no way to know just how noisy they are, since this is not reported); in some cases it may degrade the results, but in others it may show an artifactually large regression estimate driven by these outliers. In either case, the worry is not that the results are wholly explained by these noisy epochs alone, but that the results are contaminated and the magnitude of the effect may be misestimated. Removal of noisy artifacts and epochs is a robust standard in the field and for good reason – artifacts can be abundant and large in amplitude, and a claim that they do not affect the results must be substantiated. Thankfully the authors have the ability to demonstrate, not just hint, that their results are not driven or largely affected by noisy activity, and they should do so. Specifically, the authors need to report model estimates in a clean version of the dataset in which not only noisy electrodes, but also noisy epochs (movement and epileptiform artifacts) are removed. This will unequivocally demonstrate their (currently unverified) claim that noisy activity does not significantly affect their model estimates.

Related to this, in their response to point (5) below, the authors say "We formally verified that manually removing artifacts did not affect the results about PE signals observed in BGA." I'm not sure what this is referring to – if this is referring to the additional removal of noisy/epileptic electrodes, that is a different proposition than manually removing contaminated epochs. If they did carry out an analysis in which

noisy epochs were removed, this would completely address my previous point, but needs to be shown (I don't see a figure or supplementary figure that this refers to).

5. The reference to earlier papers is sufficient for comparison with healthy subjects.

6. My bad on the confusion - +RPE and -RPE should indeed refer to RPE and PPE.

7. Minor comments: all appropriately addressed in the authors' response.

Reviewer #3 (Remarks to the Author):

The authors prepared an extensively and carefully revised version. They responded thoughtfully to my concerns, which I appreciate. Upon re-reading the revised version it is much clearer to me now what is being done and what the rationale is for doing so. Removal of the 'opponent' wording clarified things significantly. Controlling for effects of learning is helpful, as is analysis of the expectation before onset of outcome, and baseline removal control. The added supplementary Figures (S3, S7) are helpful.

I can now recommend this paper for publication.

Minor issue:

1. I am assuming that subjects are not told which symbols are in the 'reward' and which in the 'punishment' category. Thus at least for the first few trials they wouldnt know (before seeing the outcome) whether this is a reward or punishment condition. It would be helpful to clarify this in results (if my interpretation is correct).

Point-by-point responses to reviewers

Reviewer #1

“The authors have thoroughly addressed all my concerns and I am happy to recommend this manuscript for publication.”

This is nice to hear!

Reviewer #2

“In their revision, the authors have mostly responded adequately to my concerns, with one important outstanding issue (see below).”

The removal of noisy electrodes and associated explanation is welcome. However, the lack of exclusion of noisy epochs is troublesome. The authors make a reasonable claim that any artifactual activity is unlikely to explain the breadth of results. However, the claim that “including all the iEEG data in the analysis, without selecting good recordings, could only play against the reported results” is false. One can imagine a situation in which a single or few trials contaminated by large artifacts could be biasing (perhaps strongly) the regression estimates. The frequency with which this happens will depend on how noisy the recordings are (and there is no way to know just how noisy they are, since this is not reported); in some cases, it may degrade the results, but in others it may show an artifactually large regression estimate driven by these outliers. In either case, the worry is not that the results are wholly explained by these noisy epochs alone, but that the results are contaminated and the magnitude of the effect may be misestimated. Removal of noisy artifacts and epochs is a robust standard in the field and for good reason – artifacts can be abundant and large in amplitude, and a claim that they do not affect the results must be substantiated. Thankfully the authors have the ability to demonstrate, not just hint, that their results are not driven or largely affected by noisy activity, and they should do so.

Specifically, the authors need to report model estimates in a clean version of the dataset in which not only noisy electrodes, but also noisy epochs (movement and epileptiform artifacts) are removed. This will unequivocally demonstrate their (currently unverified) claim that noisy activity does not significantly affect their model estimates. Related to this, in their response to point (5) below, the authors say “We formally verified that manually removing artifacts did not affect the results about PE signals observed in BGA.” I’m not sure what this is referring to – if this is referring to the additional removal of noisy/epileptic electrodes, that is a different proposition than manually removing contaminated epochs. If they did carry out an analysis in which noisy epochs were removed, this would completely address my previous point, but needs to be shown (I don’t see a figure or supplementary figure that this refers to).

We agree with the reviewer that a quantitative assessment of the effects of removing epochs contaminated with artifacts/noise was missing. We updated supplementary Figure S5 to show the regression estimates obtained after removing the pathological/noisy channels and trials (Figure S5).

We added a sentence in the methods (‘iEEG data acquisition and preprocessing’) and in the results (‘iEEG: comparison between reward and punishment PE’) to explain how trials with artifacts were excluded.

Methods

We also excluded recording sites that were part of the epileptogenic zone by identifying with the neurologists (PK and SR) all recording sites involved at seizure onset and/or sites that were located within the cortical resection (if any) performed after the sEEG;

furthermore, trials during which iEEG activity was higher or lower than four times the standard deviation of the average signal were excluded.

Overall, we excluded 8.7 % of trials (range between ROIs: 5.5 to 10.8 %). We also visually inspected the removed trials as a sanity check.

Results

To confirm this dissociation between reward PE and punishment PE signaling regions, we performed a number of control analyses, at different levels from iEEG data preprocessing to model-based regressions. First, the functional dissociation was unchanged when using an alternative procedure for extracting BGA (See Methods and Figure S4) or after removing recording sites with pathological activity and trials with artifacts (See Methods and Figure S5).

Figure S5. Dissociation of reward PE (R-Qr) and punishment PE (P-Qp) signals after excluding channels and trials contaminated with pathological activity and/or artifacts. Time course of regression estimates obtained from linear fit of BGA with PE modeled separately for the reward (blue) and punishment (red) conditions (PPE: punishment prediction error; RPE: reward prediction error). Horizontal bold lines indicate significant difference between conditions (blue: RPE>PPE; red: PPE>RPE; $p < 0.05$). Shaded areas represent inter-sites SEM.

Reviewer #3

The authors prepared an extensively and carefully revised version. They responded thoughtfully to my concerns, which I appreciate. I can now recommend this paper for publication.

Thank you for the supportive comments.

Minor issue:

1. I am assuming that subjects are not told which symbols are in the 'reward' and which in the 'punishment' category. Thus at least for the first few trials they wouldn't know (before seeing the outcome) whether this is a reward or punishment condition. It would be helpful to clarify this in results (if my interpretation is correct).

We added a sentence in the result section to clarify this point.

Patients were instructed to do their best to maximize the monetary gains and to minimize the monetary losses during the task. No further information was given regarding the exact task structure.

REVIEWERS' COMMENTS

Reviewer #2 (Remarks to the Author):

The authors have now fully responded to my concerns and I'm happy to recommend this paper for publication.